# SciVideoBench: Benchmarking Scientific Video Reasoning in Large Multimodal Models

## Abstract

**Abstract:** Large Multimodal Models (LMMs) have achieved remarkable progress across various capabilities; however, complex video reasoning in the scientific domain remains a significant and challenging frontier. Current video benchmarks predominantly target general scenarios where perception/recognition is heavily relied on, while with relatively simple reasoning tasks, leading to saturation and thus failing to effectively evaluate advanced multimodal cognitive skills. To address this critical gap, we introduce SciVideoBench, a rigorous benchmark specifically designed to assess advanced video reasoning in scientific contexts. SciVideoBench consists of 1,000 carefully crafted multiple-choice questions derived from cutting-edge scientific experimental videos spanning over 25 specialized academic subjects and verified by a semi-automatic system. Each question demands sophisticated domain-specific knowledge, precise spatiotemporal perception, and intricate logical reasoning, effectively challenging models' higher-order cognitive abilities. Our evaluation highlights significant performance deficits in state-of-the-art proprietary and open-source LMMs, including Gemini 2.5 Pro and Qwen2.5-VL, indicating substantial room for advancement in video reasoning capabilities. Detailed analyses of critical factors such as reasoning complexity and visual grounding provide valuable insights and clear direction for future developments in LMMs, driving the evolution of truly capable multimodal AI co-scientists. We hope SciVideoBench could fit the interests of the community and help to push the boundary of cutting-edge AI for border science.

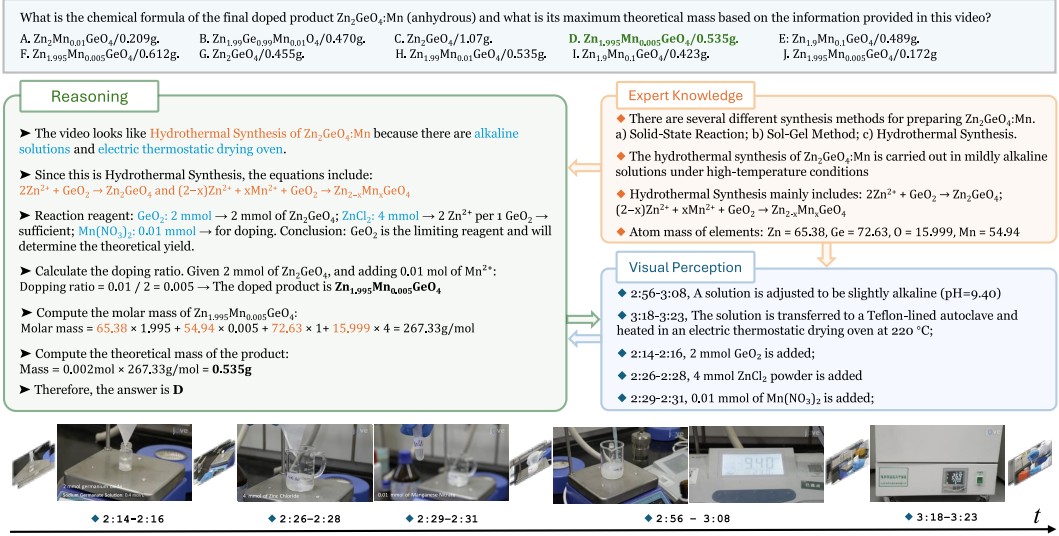

Figure 1: SciVideoBench features *research-level* experimental videos accompanied by challenging questions that rigorously evaluate advanced video understanding. It emphasizes the *synergistic interaction* among accurate visual perception, expert knowledge, and logical reasoning.

# 1 INTRODUCTION

Large Multimodal Models (LMMs) (OpenAI, 2024; Anil et al., 2023; Lu et al., 2024; Wang et al., 2024) have demonstrated rapid advancements across a diverse range of capabilities, including conversational interaction (Hendrycks et al., 2020; Yang et al., 2018), code generation (Jiménez et al., 2023; Dong et al., 2024), mathematical reasoning (Cobbe et al., 2021; Zhao et al., 2025a), and image understanding (Yue et al., 2024; Hudson & Zitnick, 2019). Among the various input modalities, video presents a particularly rich and complex form of multimodal data, uniquely integrating temporal dynamics, spatial perception, and high-level semantic reasoning. Consequently, video understanding has emerged as a critical frontier, pivotal for advancing LMMs towards next-generation applications in fields such as robotics, interactive education, and scientific discovery.

To evaluate LMM performance, numerous video benchmarks have been developed (Fang et al., 2024; Fu et al., 2024; Hu et al., 2025; Wu et al., 2024b; Pătrăucean et al., 2023; Zhou et al., 2018; Xiao et al., 2021; Song et al., 2024; Xiao et al., 2024; Li et al., 2024c; Nagrani et al., 2024), most of which focus on general domains such as movies (Fu et al., 2024; Song et al., 2024), daily activities (Mangalam et al., 2023; Xiao et al., 2021), and instructional content (Zhou et al., 2018). While these benchmarks once posed significant challenges, state-of-the-art models like Gemini 2.5 Pro (Google, 2025) now achieve saturated performance, exceeding 85% accuracy on widely used datasets such as VideoMME (Fu et al., 2024) and Neptune (Nagrani et al., 2024). To move beyond these general domains, more recent initiatives (Hu et al., 2025; Song et al., 2025; Zhao et al., 2025b) have introduced scientific and educational videos to assess deeper domain knowledge and reasoning. However, most of the reasoning in these datasets remains at a **college-level**, relying largely on classical theory and textbook knowledge. In many cases, success can be achieved either through visual recognition combined with memorized knowledge or through reasoning that does not depend on the video at all, thereby lacking genuine multimodal reasoning. For instance, MMVU (Zhao et al., 2025b) contains questions with visually irrelevant hypotheses that can be answered without watching the video. As a result, these benchmarks pose only limited challenges: proprietary LMMs already perform strongly, and even language-based reasoning frameworks (Zhang et al., 2025; Yu et al., 2025) achieve competitive results.

To address this gap beyond college-level reasoning, we introduce SCIVIDEOBENCH, a benchmark specifically designed to rigorously evaluate advanced video reasoning at the **research-level**, where cutting-edge scientific problems and theories are investigated. SCIVIDEOBENCH consists of 1,000 carefully constructed multiple-choice questions developed through a multi-stage agent–human collaborative pipeline and derived from experimental videos published alongside peer-reviewed journal articles. Each question is categorized as conceptual, hypothetical, or quantitative, ensuring coverage of diverse reasoning skills. As illustrated in Figure 1, SCIVIDEOBENCH challenges models not only to achieve precise spatio-temporal grounding but also to demonstrate deeper domain knowledge and perform sophisticated logical reasoning, thereby providing a more realistic and demanding testbed for scientific video understanding.

Our evaluation of the most recent LMMs on SCIVIDEOBENCH reveals consistently low accuracy, underscoring the significant challenge this new benchmark presents and, consequently, the substantial opportunity for advancing research-level video reasoning capabilities. In-depth analysis of the results reveals that factors such as model architecture, reasoning capacity, and perceptual grounding play a critical role in shaping video reasoning performance. These insights not only offer clear guidance for future research efforts aimed at developing more sophisticated LMMs but also emphasize the broader potential of SCIVIDEOBENCH. As the first comprehensive research-level video reasoning benchmark, SCIVIDEOBENCH not only provides a rigorous testbed for evaluating current video reasoning abilities, but also serves as a catalyst for innovation, fostering the development of highly capable AI co-scientists that can accelerate future scientific discovery.

# 2 RELATED WORK

## 2.1 VIDEO REASONING BENCHMARKS

Recent benchmarks have increasingly emphasized complex temporal reasoning and multimodal understanding to evaluate LMMs beyond the traditional video QA benchmark paradigm (Yu et al., 2019; Mangalam et al., 2023; Xu et al., 2017; Xiao et al., 2021; Wu et al., 2024a). Perception

Table 1: Benchmark comparison for video understanding and reasoning tasks. Compared with college-level questions that focus on classic theories or textbook content, research-level questions demand deeper domain knowledge and reasoning grounded in real, cutting-edge research scenarios.

| Dataset | Video Domain | Difficulty | Knowledge Driven | Reasoning Intensive | Avg. Duration (s) | Question Type |
|---|---|---|---|---|---|---|
| MovieChat-1K (Song et al., 2024) | Movie | Elementary | ✗ | ✗ | 564 | Open-ended |
| MLVU (Xiao et al., 2024) | General | Elementary | ✗ | ✗ | 930 | Multi-choice |
| MVBench (Li et al., 2024c) | General | Elementary | ✗ | ✗ | 16 | Multi-choice |
| LongVideoBench (Wu et al., 2024b) | General | Elementary | ✗ | ✗ | 473 | Multi-choice |
| TempCompass (Liu et al., 2024) | General | Elementary | ✗ | ✗ | < 30 | Multi-choice |
| Video-MME (Fu et al., 2024) | General | Elementary | ✗ | ✗ | 1018 | Multi-choice |
| VSI-Bench (Yang et al., 2024b) | Embodied | Elementary | ✗ | ✗ | 122 | Multi-choice |
| MMWorld (He et al., 2024) | General | Elementary | ✓ | ✗ | 107 | Multi-choice |
| MMVU (Zhao et al., 2025b) | Scientific | College | ✓ | ✗ | 51 | Multi-choice |
| Video-MMMU (Hu et al., 2025) | Lecture | College | ✓ | ✓ | 506 | Multi-choice |
| Video-MMLU (Song et al., 2025) | Lecture | College | ✓ | ✓ | 109 | Open-ended |
| **SciVideoBench (ours)** | Scientific | Research | ✓ | ✓ | 484 | Multi-choice |

Test (Pătrăucean et al., 2023) examines multiple reasoning modes such as descriptive, predictive, and counterfactual reasoning in real-world scenarios. MVBench (Li et al., 2024c) proposes 20 temporally grounded challenges that require understanding actions, object interactions, and motion cues across multiple frames. Video-MME (Fu et al., 2024) considers subtitle and audio streams to promote multi-source comprehension. MMBench-Video targets long-context temporal reasoning by leveraging hour-long videos with open-ended QA. Domain-specific benchmarks have also emerged. WorldQA (Zhang et al., 2024c) focuses on long-chain reasoning with multimodal world knowledge, Video-MMMU (Hu et al., 2025) collects expert-level instructional videos across disciplines for multi-stage knowledge acquisition, and Video-MMLU (Song et al., 2025) centers on lecture-level understanding in math, physics, and chemistry. Our work extends this line of research by introducing a scientific video reasoning benchmark grounded in real experimental settings, emphasizing measurement, calculation, and conceptual reasoning involved in specific experiments.

## 2.2 AI FOR SCIENCE

Artificial intelligence has recently driven major advances across biology (Jumper et al., 2021; Yang et al., 2023; Lin et al., 2022), chemistry (Qiao et al., 2020; Bran et al., 2023), materials science (Kim et al., 2023; Chen & Ong, 2023), and mathematics (Lewkowycz et al., 2022), achieving performance that in many cases rivals or surpasses traditional methods. AlphaFold (Jumper et al., 2021; Yang et al., 2023) revolutionized protein structure prediction with near-experimental accuracy, GNoME (Kim et al., 2023) discovered over 700,000 stable crystal structures via graph neural networks, and models like OrbNet (Qiao et al., 2020) and ChemCrow (Bran et al., 2023) combine quantum chemistry with language models for simulation and synthesis planning. Yet despite these breakthroughs, AI remains far from acting as a versatile researcher capable of complex, expert-level tasks across domains. To this end, we introduce SCIVIDEOBENCH, a benchmark for scientific video reasoning that demands visual perception, domain knowledge, and intricate reasoning, aiming to assess the gap between current LMMs and expert-level scientific reasoning while fostering the development of next-generation AI for science.

## 3 DATASET CONSTRUCTION

### 3.1 VIDEO COLLECTION

To construct a high-quality benchmark for advanced scientific reasoning, we collect 241 research-grade experimental videos from the *Journal of Visualized Experiments* (JoVE)[1] , a peer-reviewed platform publishing methodological videos across diverse scientific disciplines. These professionally produced and narratively structured videos clearly demonstrate laboratory protocols, scien-

---

[1]https://www.jove.com

tific phenomena, and technical instrumentation, making them an ideal foundation for a benchmark grounded in authentic scientific practice. Each video is accompanied by a peer-reviewed manuscript and synchronized audio narration: the manuscript details experimental protocols and results, while the narration provides temporally aligned explanations of each step as it unfolds. This tri-modal alignment—**video**, **audio**, and **text**—supports principled question generation and rigorous answer verification, ensuring that questions are both visually grounded and scientifically meaningful. We focus on four foundational domains—**physics**, **chemistry**, **biology**, and **medicine**—covering a wide spectrum of procedural complexity and reasoning challenges. Videos are selected to include measurable variables (e.g., reaction time, temperature, applied force), observable causal relationships, and logical experimental sequences, thereby enabling *conceptual*, *hypothetical*, and *quantitative* reasoning. This targeted curation ensures that each video in SCIVIDEOBENCH provides rich multimodal cues essential for rigorous scientific reasoning and serves as an ideal testbed for evaluating LMMs.

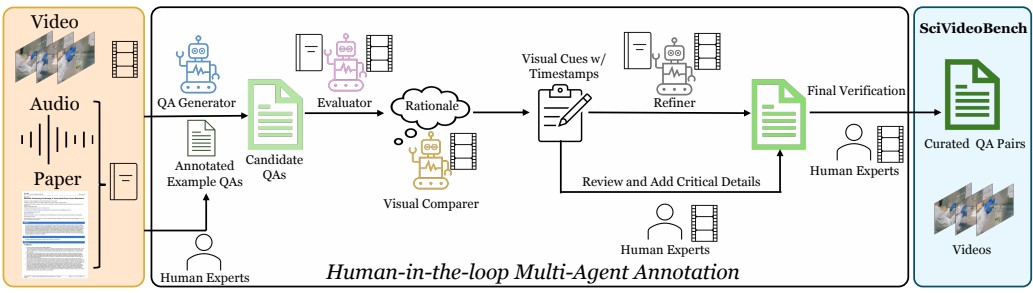

Figure 2: Overview of our annotation pipeline. We manually annotate example QA pairs, and then a multi-agent LLM system generates and refines QA pairs for the rest videos: the QA Generator produces initial questions, the Evaluator answers them with reasoning, the Visual Comparer checks for visual grounding and timestamps cues, and the Refiner ensures questions rely on video content and improves option quality. Human experts verify and refine the final QA pairs.

## 3.2 ANNOTATION PIPELINE

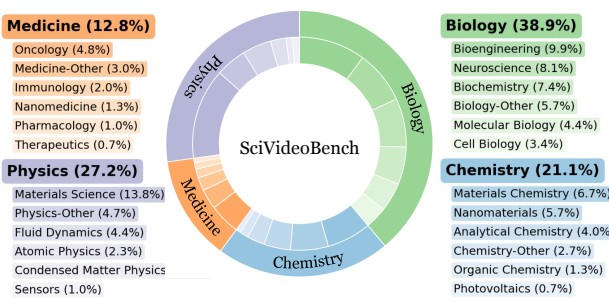

Figure 3: Discipline and subject distribution.

To generate high-quality, research-level QA pairs, we design a semi-automatic annotation pipeline (Figure 2) that integrates video, aligned transcripts, and paired research papers to provide rich scientific and procedural context. We begin with a small set of exemplar cases: four human experts (one PhD student per subject) review representative papers and their associated videos to extract key concepts and procedural steps, identify corresponding visual cues, and construct sophisticated QA pairs. These are refined and expanded via GPT-4o into 12 high-quality examples per reasoning type, which serve as prompts for a human-in-the-loop multi-agent annotation system. In the second stage, distinct Gemini 2.5 Pro agents assume specialized roles: the **QA Generator** produces candidate questions using multimodal inputs; the **Evaluator** answers them with a rationale; the **Visual Comparer** verifies grounding in the video and provides precise timestamps; and the **Refiner** strengthens visual dependence and adjusts distractors. Human verifiers then review these outputs, checking timestamps and ensuring answerability based on visual evidence; when critical details (e.g., mass or temperature) appear only in transcripts, they trigger scripts to overlay this information onto relevant frames. Before the final human review, ==we introduce an additional GPT-4o-based filtering stage to further ensure visual dependence and eliminate annotation artifacts. Specifically, GPT-4o performs two complementary checks: (1) an *option-only test* to detect answer-pattern leakage or distractor weaknesses, and (2) a *visual-blind test* in which GPT-4o answers using only the question and options without any video input. Items that GPT-4o answers==

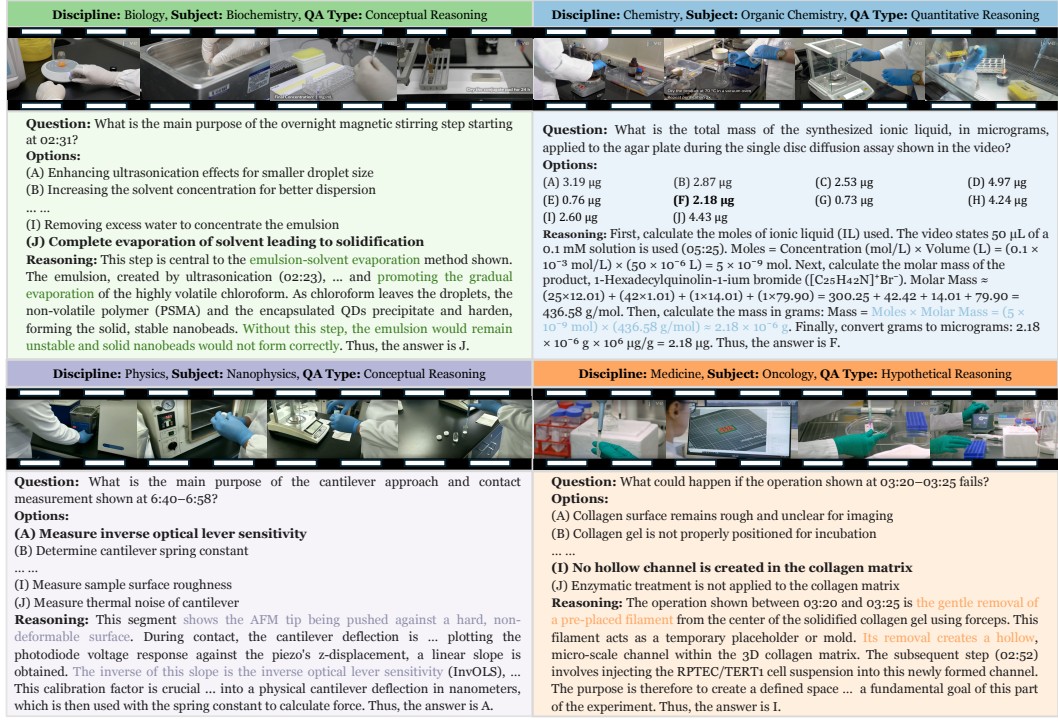

Figure 4: Examples of SCIVIDEOBENCH.

correctly in either setting are flagged as insufficiently grounded and are revised or discarded. This automated audit reliably removes questions solvable through textual priors or stylistic biases before they reach the human verification stage. Finally, experts manually review and correct errors, ensuring that each question is visually grounded, scientifically accurate, and aligned with the experiment's core contribution. Consequently, more than 3,000 preliminary questions were removed during refinement, yielding a final set of 1,000 rigorously verified items. Detailed annotation process, the agent prompts and metadata examples are provided in Section C in the Appendix.

## 3.3 STATISTICS

We collected a total of 241 experimental videos spanning four major domains and covering more than 25 distinct scientific subjects, as illustrated in Figure 3. The average video duration is 484 seconds, which ensures that the benchmark reflects the complexity and extended reasoning often required in real-world scientific experiments. Building upon these videos, we annotated a total of 1,000 challenging questions that demand research-level knowledge for both perception and reasoning. To further capture the nature of academic research and experimental analysis, we carefully designed three distinct question types (*conceptual*, *hypothetical*, and *quantitative*) that reflect common reasoning scenarios observed across the videos, as illustrated in Figure 4. More details of the statistics and question type can be found in Section D in the Appendix.

## 4 EXPERIMENTS

In this section, we systematically evaluate existing LLMs on SCIVIDEOBENCH across multiple dimensions to highlight their strengths and limitations. Specifically, we compare proprietary and open-source models, analyze the impact of chain-of-thought prompting, and examine how model size and LLM backbone capacity affect performance. Additional analyses of audio input, different reasoning types, and disciplines are provided in Section G, H, I in the Appendix.

## 4.1 MODELS

To investigate the importance of visual information in our benchmark, we first conduct the vision-blind evaluation using GPT-4o (OpenAI, 2024) and Qwen2.5 series (Yang et al., 2024a). Further, we evaluate five proprietary LMMs that have superior performance over other popular video benchmarks, including Gemini-1.5-Pro (Anil et al., 2023), Gemini-2.0-Flash (Anil et al., 2023), Gemini-2.5-Pro (Google, 2025), and GPT-4o (OpenAI, 2024). We also comprehensively evaluate 30 open-sourced LMMs with the parameter volume ranging from as tiny as 0.5B to 78B, including Qwen-VL series (Bai et al., 2025; Wang et al., 2024), InternVL series (Chen et al., 2024b; Zhu et al., 2025; Chen et al., 2024a), InternVide2.5-Chat-8B (Wang et al., 2025), LLaVA-OneVision Series (Li et al., 2024a), LLaVA-NeXT-Video-32B series (Li et al., 2024b), and LongVA (Zhang et al., 2024b).

## 4.2 EVALUATION SETTING

We adopt the default frame sampling strategy for all open-source models. Specifically, we sample 768 frames for Qwen-2.5-VL (Bai et al., 2025), 512 frames for InternVideo2.5-Chat-8B (Wang et al., 2025), 128 frames for LongVA (Zhang et al., 2024b), 16 frames for InternVL2 (Chen et al., 2024b), and 32 frames for InternVL3 (Zhu et al., 2025), LLaVA-OneVision (Li et al., 2024a), and LLaVA-NeXT-Video-32B (Li et al., 2024b). For Gemini (Anil et al., 2023; Comanici et al., 2025; Google, 2025), we set the frame rate to 1 FPS, and for GPT-4o (OpenAI, 2024), we use 256 frames. The temperature is fixed to 0 to ensure stable and reproducible evaluation results. All evaluations are using the LMM-Eval toolkit (Zhang et al., 2024a) and are conducted on 8 NVIDIA H100 GPUs.

Further, we conduct a human evaluation by recruiting three graduate students from each discipline to answer questions in a closed-book setting. Their overall accuracy is only 17.4% (Table 2), showing that even advanced students cannot handle the benchmark, which requires research-level expertise.

## 4.3 EVALUATION ANALYSIS

**Blind Baseline Results** To assess the role of visual input in SCIVIDEOBENCH, we evaluate blind baselines that use only textual content. GPT-4o achieves just 15.8% overall accuracy, with Quantitative Reasoning near random (11.8%), while Qwen2.5 models range from chance-level in small variants (0.5B–3B) to only 18.9% at 72B. Even with scaling, none surpass 20% accuracy without video, and the presence of time-specific, observation-heavy cues (e.g., "starting at 02:31") further confirms that visual information is indispensable for solving SCIVIDEOBENCH.

**Proprietary vs. Open-Source Models** As shown in Table 2, proprietary models substantially outperform open-source counterparts on our SCIVIDEOBENCH. The strongest proprietary system, Gemini-2.5-Pro, achieves 64.30% overall accuracy, whereas the best open-source model, InternVL-3-78B-Instruct, reaches only 38.80%. The gap is especially pronounced in Quantitative Reasoning, where Gemini-2.5-Pro attains 50.61%—more than double the best open-source score (22.04% by InternVL2-Llama3-76B). Nevertheless, top open-source models demonstrate competitiveness against earlier proprietary models: for example, most InternVL-3 variants outperform Gemini-1.5-Pro, and even InternVL-3-2B surpasses it on Conceptual Reasoning. At the lower end, performance drops sharply. The weakest open-source model, LLaVA-OneVision-0.5B, achieves just 12.10% overall, with its Quantitative score (5.71%) falling below even the blind GPT-4o baseline (11.84%).

> **Takeaway: Proprietary vs. Open-Source Models**
>
> Gemini-2.5-Pro leads overall. Proprietary models dominate quantitative reasoning, while top open-source models remain competitive on conceptual/hypothetical tasks.

**The Impact of Chain-of-Thought Prompt** When prompting with chain-of-thought, Gemini-1.5-Pro shows the largest gain, with overall accuracy rising by +21.1% and Quantitative Reasoning by +25.3%, surpassing even Gemini-2.5-Pro; Gemini-2.0-Flash and GPT-4o improve by +14.0% and +10.1%, respectively, with GPT-4o's Quantitative score increasing from 11.8% to 34.3%, underscoring the transformative effect of explicit reasoning steps. On average, Quantitative tasks benefit most (+21.8%), while Conceptual gains are more modest (+12.5%), suggesting that CoT especially enhances multi-step numerical reasoning. In contrast, open-source models often suffer slight drops in overall accuracy under CoT, yet their Quantitative performance improves markedly. For instance, Qwen2.5-VL-32B-Instruct drops 0.7% overall but improves from 11.4% to 18.0% (+57.1% relative)

Table 2: Evaluation Results of Proprietary Models and Open-Source Models on SCIVIDEOBENCH.

| Models | LLM | Size | Overall | Question Type | | | Discipline | | | |
|---|---|---|---|---|---|---|---|---|---|---|
| | | | | Conceptual | Hypothetical | Quantitative | Biology | Chemistry | Medicine | Physics |
| *Random Guess* | | | | | | | | | | |
| - | - | - | 10.00 | 10.00 | 10.00 | 10.00 | 10.00 | 10.00 | 10.00 | 10.00 |
| *Human Evaluation (Graduate Students)* | | | | | | | | | | |
| - | - | - | 17.40 | 18.11 | 18.70 | 14.29 | 15.88 | 16.06 | 21.19 | 18.88 |
| *Vision-Blind Baselines* | | | | | | | | | | |
| GPT-4o (OpenAI, 2024) | - | - | 15.80 | 14.05 | 20.00 | 11.84 | 17.53 | 13.95 | 16.13 | 14.33 |
| | - | 0.5B | 12.40 | 11.35 | 13.51 | 12.24 | 13.20 | 13.94 | 11.21 | 10.97 |
| | - | 1.5B | 13.40 | 11.62 | 19.22 | 6.94 | 14.43 | 12.73 | 14.95 | 11.91 |
| | - | 3B | 16.40 | 17.03 | 20.52 | 8.98 | 15.89 | 16.36 | 15.89 | 17.24 |
| Qwen2.5 (Yang et al., 2024a) | - | 7B | 16.70 | 18.65 | 19.74 | 8.98 | 18.34 | 10.91 | 14.95 | 18.18 |
| | - | 32B | 17.10 | 18.11 | 21.30 | 8.98 | 19.32 | 16.97 | 13.08 | 15.67 |
| | - | 72B | 18.90 | 21.89 | 18.44 | 15.10 | 19.07 | 18.79 | 20.56 | 18.18 |
| *Proprietary Models w/ Direct Answer* | | | | | | | | | | |
| Gemini-2.5-Pro (Google, 2025) | - | - | **64.30** | **69.73** | **67.79** | 50.61 | **64.79** | **61.82** | **74.77** | **61.44** |
| Gemini-2.5-Flash (Comanici et al., 2025) | - | - | 46.40 | 50.81 | 44.16 | 43.27 | 44.01 | 49.70 | 55.14 | 44.83 |
| Gemini-1.5-Pro (Anil et al., 2023) | - | - | 27.50 | 27.84 | 28.31 | 25.71 | 27.38 | 26.06 | 27.10 | 28.53 |
| Gemini-2.0-Flash (Anil et al., 2023) | - | - | 25.70 | 28.38 | 24.94 | 22.86 | 24.69 | 26.06 | 22.43 | 27.90 |
| GPT-4o (OpenAI, 2024) | - | - | 24.90 | 30.27 | 28.05 | 11.84 | 21.52 | 29.70 | 31.78 | 24.45 |
| *Proprietary Models w/ Chain-of-Thought Prompt* | | | | | | | | | | |
| Gemini-1.5-Pro (Anil et al., 2023) | - | - | 48.60(+21.10) | 47.03(+19.19) | 48.57(+20.26) | 51.02(+25.31) | 52.60(+25.22) | 44.19(+18.13) | 54.84(+27.74) | 44.78(+16.25) |
| Gemini-2.0-Flash (Anil et al., 2023) | - | - | 39.70(+14.00) | 39.19(+10.81) | 39.74(+14.80) | 40.41(+17.55) | 41.56(+16.87) | 40.70(+14.64) | 35.48(+13.05) | 37.01(+9.11) |
| GPT-4o (OpenAI, 2024) | - | - | 35.00(+10.10) | 37.84(+7.57) | 32.73(+4.68) | 34.29(+22.45) | 31.78(+10.26) | 37.58(+7.88) | 36.45(+4.67) | 37.30(+12.85) |
| *Open-Source Models w/ Chain-of-Thought Prompt* | | | | | | | | | | |
| InternVL-3-78B (Zhu et al., 2025) | Qwen2.5 | 72B | 37.90 | 51.62 | 35.58 | 20.82 | 38.39 | 36.36 | 38.32 | 37.93 |
| InternVL-3-14B (Zhu et al., 2025) | Qwen2.5 | 14B | 34.20 | 46.49 | 30.65 | 21.22 | 35.70 | 30.30 | 37.38 | 33.23 |
| InternVL-3-14B-Instruct (Zhu et al., 2025) | Qwen2.5 | 14B | 31.50 | 42.43 | 27.79 | 20.82 | 30.07 | 32.73 | 37.38 | 30.72 |
| InternVL-3-8B (Zhu et al., 2025) | Qwen2.5 | 7B | 25.50 | 34.32 | 24.16 | 14.29 | 25.18 | 24.85 | 28.97 | 25.08 |
| InternVL2-Llama3-76B (Chen et al., 2024b) | Hermes2 | 70B | 24.90 | 27.30 | 22.34 | 25.31 | 25.18 | 24.24 | 28.97 | 23.51 |
| InternVL-3-9B-Instruct (Zhu et al., 2025) | InternLM3 | 8B | 24.00 | 29.19 | 22.08 | 19.18 | 25.92 | 27.88 | 17.76 | 21.63 |
| InternVL-3-2B (Zhu et al., 2025) | Qwen2.5 | 1.5B | 22.20 | 30.00 | 21.82 | 11.02 | 21.76 | 16.36 | 25.23 | 24.76 |
| Qwen2.5-VL-32B-Instruct (Bai et al., 2025) | Qwen2.5 | 32B | 20.80 | 20.54 | 22.86 | 17.96 | 20.78 | 18.18 | 16.82 | 23.51 |
| LLaVA-OneVision-7B (Li et al., 2024a) | Qwen2.5 | 7B | 19.90 | 24.32 | 20.52 | 12.24 | 16.87 | 21.21 | 28.04 | 20.38 |
| Qwen2.5-VL-3B-Instruct (Bai et al., 2025) | Qwen2.5 | 3B | 18.10 | 19.19 | 18.96 | 15.10 | 17.60 | 18.79 | 19.63 | 17.87 |
| InternVL-3-1B (Zhu et al., 2025) | Qwen2.5 | 0.5B | 14.00 | 16.76 | 13.51 | 10.61 | 15.40 | 13.94 | 14.02 | 12.23 |
| LLaVA-OneVision-0.5B (Li et al., 2024a) | Qwen2.5 | 0.5B | 10.40 | 11.08 | 10.65 | 8.98 | 8.56 | 12.12 | 13.08 | 10.97 |
| *Open-Source Models w/ Direct Answer (0.5B - 4B)* | | | | | | | | | | |
| InternVL-3-2B-Instruct (Zhu et al., 2025) | Qwen2.5 | 1.5B | 24.00 | 31.08 | 23.90 | 13.47 | 24.21 | 21.82 | 23.36 | 25.08 |
| InternVL-3-2B (Zhu et al., 2025) | Qwen2.5 | 1.5B | 22.90 | 31.08 | 22.60 | 11.02 | 21.52 | 21.21 | 24.30 | 25.08 |
| Qwen2.5-VL-3B-Instruct (Bai et al., 2025) | Qwen2.5 | 3B | 22.50 | 32.43 | 20.78 | 10.20 | 18.61 | 21.76 | 22.03 | 28.67 |
| InternVL2-4B (Chen et al., 2024b) | Phi-3-mini | 4B | 21.30 | 26.22 | 24.16 | 9.39 | 18.09 | 24.85 | 30.84 | 20.38 |
| InternVL-3-1B-Instruct (Zhu et al., 2025) | Qwen2.5 | 0.5B | 18.90 | 26.76 | 18.44 | 7.76 | 18.34 | 18.18 | 22.43 | 18.81 |
| InternVL-3-1B (Zhu et al., 2025) | Qwen2.5 | 0.5B | 18.50 | 25.95 | 18.44 | 7.35 | 17.60 | 16.36 | 25.23 | 18.50 |
| InternVL2-1B (Chen et al., 2024b) | InternLM2 | 0.5B | 14.40 | 17.57 | 14.03 | 10.20 | 12.96 | 13.94 | 16.82 | 15.67 |
| InternVL2-2B (Chen et al., 2024b) | InternLM2 | 2B | 13.10 | 14.59 | 15.06 | 7.76 | 11.98 | 12.12 | 12.15 | 15.36 |
| LLaVA-OneVision-0.5B (Li et al., 2024a) | Qwen2.5 | 0.5B | 12.10 | 15.41 | 12.99 | 5.71 | 11.74 | 13.94 | 13.08 | 11.29 |
| *Open-Source Models w/ Direct Answer (7B - 14B)* | | | | | | | | | | |
| InternVL-3-14B (Zhu et al., 2025) | Qwen2.5 | 14B | 35.70 | 53.51 | 35.32 | 9.39 | 35.94 | 33.94 | 38.32 | 35.42 |
| InternVL-3-14B-Instruct (Zhu et al., 2025) | Qwen2.5 | 14B | 35.70 | 53.24 | 36.36 | 8.16 | 35.70 | 34.55 | 38.32 | 35.42 |
| InternVL-3-8B (Zhu et al., 2025) | Qwen2.5 | 7B | 30.50 | 44.59 | 30.39 | 9.39 | 29.10 | 31.52 | 35.51 | 30.09 |
| InternVL-3-8B-Instruct (Zhu et al., 2025) | Qwen2.5 | 7B | 29.40 | 43.78 | 29.35 | 7.76 | 26.16 | 31.52 | 34.58 | 30.72 |
| InternVL-3-9B-Instruct (Zhu et al., 2025) | InternLM3 | 8B | 29.20 | 40.27 | 30.91 | 9.80 | 29.10 | 29.70 | 33.64 | 27.69 |
| InternVL-3-9B (Zhu et al., 2025) | InternLM3 | 8B | 27.20 | 38.65 | 27.79 | 8.98 | 25.67 | 26.06 | 31.78 | 28.21 |
| Qwen2.5-VL-7B-Instruct (Bai et al., 2025) | Qwen2.5 | 7B | 26.30 | 35.95 | 28.05 | 8.98 | 22.56 | 22.28 | 28.81 | 29.02 |
| InternVideo2.5-Chat-8B (Wang et al., 2025) | Qwen2.5 | 7B | 25.30 | 37.84 | 23.12 | 9.80 | 20.29 | 23.64 | 31.78 | 30.41 |
| InternVL2-8B (Chen et al., 2024b) | InternLM2 | 7B | 19.40 | 24.86 | 18.96 | 11.84 | 17.85 | 21.21 | 19.63 | 20.38 |
| LLaVA-OneVision-7B (Li et al., 2024a) | Qwen2.5 | 7B | 18.80 | 23.51 | 19.22 | 11.02 | 15.56 | 23.03 | 26.17 | 18.18 |
| LongVA (Zhang et al., 2024b) | Qwen2 | 7B | 14.30 | 16.15 | 14.94 | 14.94 | 13.69 | 15.00 | 15.65 | 14.11 |
| *Open-Source Models w/ Direct Answer (26B - 40B)* | | | | | | | | | | |
| InternVL-3-38B (Zhu et al., 2025) | Qwen2.5 | 32B | 38.30 | 53.78 | 38.44 | 14.69 | 36.67 | 40.00 | 42.06 | 38.24 |
| InternVL-3-38B-Instruct (Zhu et al., 2025) | Qwen2.5 | 32B | 37.30 | 52.43 | 37.14 | 14.69 | 35.94 | 39.39 | 40.19 | 36.99 |
| InternVL2-40B (Chen et al., 2024b) | Hermes2 | 34B | 23.80 | 28.38 | 23.64 | 17.14 | 22.74 | 21.82 | 30.84 | 23.82 |
| Qwen2.5-VL-32B-Instruct (Bai et al., 2025) | Qwen2.5 | 32B | 21.50 | 24.86 | 24.68 | 11.43 | 22.49 | 16.36 | 20.56 | 23.20 |
| LLaVA-NeXT-Video-32B (Li et al., 2024b) | Qwen2 | 32B | 21.10 | 26.22 | 22.86 | 10.61 | 19.80 | 22.42 | 23.36 | 21.32 |
| InternVL2-26B (Chen et al., 2024b) | InternLM2 | 20B | 19.50 | 21.89 | 20.26 | 14.69 | 18.83 | 18.18 | 22.43 | 20.06 |
| *Open-Source Models w/ Direct Answer (> 70B)* | | | | | | | | | | |
| InternVL-3-78B-Instruct (Zhu et al., 2025) | Qwen2.5 | 72B | 38.80 | 57.30 | 39.74 | 9.39 | 37.90 | 39.39 | 46.73 | 36.99 |
| InternVL-3-78B (Zhu et al., 2025) | Qwen2.5 | 72B | 38.50 | 56.76 | 39.22 | 9.80 | 37.65 | 37.58 | 46.73 | 37.30 |
| InternVL2-Llama3-76B (Chen et al., 2024b) | Hermes2 | 70B | 26.30 | 28.38 | 27.01 | 22.04 | 24.94 | 29.70 | 29.91 | 25.08 |
| Qwen2.5-VL-72B-Instruct (Bai et al., 2025) | Qwen2.5 | 72B | 20.30 | 21.62 | 20.78 | 17.55 | 21.27 | 16.97 | 24.30 | 19.44 |

in Quantitative. This reflects a trade-off: open-source models are less robust in producing faithful causal explanations and may hallucinate or over-elaborate in conceptual or hypothetical reasoning, but they benefit when CoT scaffolds arithmetic and structured comparisons. Detailed analysis and examples can be found in Section J in the Appendix. Crucially, the gains on SCIVIDEOBENCH far exceed those reported on existing video reasoning benchmarks—for Gemini-2.0-Flash, CoT improves only +3.0% on MMVU, +12.5% on Video-Holmes, +6.6% on VideoMathQA, and +2.4% on MMR-V—whereas on SCIVIDEOBENCH the improvement is +14.0%. This significant gap demonstrates that **SCIVIDEOBENCH inherently demands more complex, multi-step reasoning than prior benchmarks, establishing it as a more challenging and discriminative testbed for evaluating multimodal reasoning capabilities**.

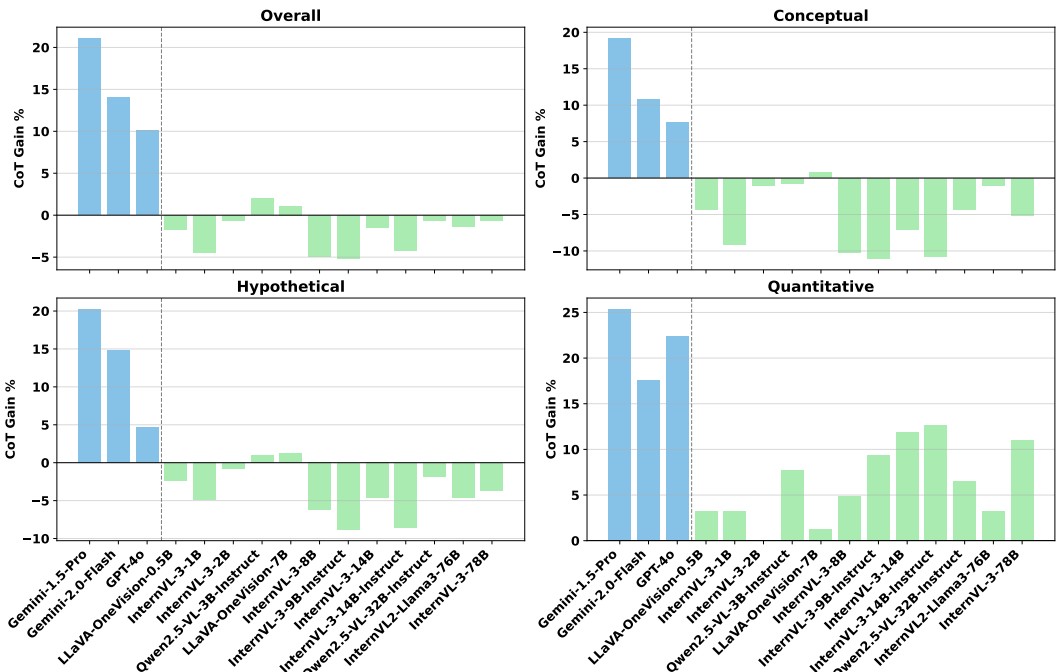

Figure 5: Chain-of-thought performance gains across proprietary and open-source models. Proprietary models exhibit consistent improvements across all reasoning types, whereas open-source models benefit primarily in quantitative reasoning but show declines in conceptual and hypothetical reasoning. This contrast not only underscores that the quantitative tasks in SCIVIDEOBENCH demand sophisticated multi-step reasoning well supported by chain-of-thought prompting, but also highlights the robustness gap between proprietary models and open-source models when prompted by chain-of-thought. Best viewed in color.

> **Takeaway: The Impact of Chain-of-Thought Prompt**
>
> Chain-of-thought prompting consistently boosts quantitative reasoning for both proprietary and open-source models, while improvements in conceptual and hypothetical reasoning are observed almost only in proprietary models.

**The Impact of Model Scaling** We first examine the effect of scaling LMMs within the same series. In general, increasing model size improves accuracy, but the trend is neither linear nor guaranteed. For InternVL-3, accuracy rises substantially from 14.0% to 35.7%, with the largest gain from 1B to 7B (+11%), particularly in Conceptual reasoning (16.8% $rightarrow$ 43.8%), though the 9B model (27.2%) underperforms the smaller 8B-Instruct (29.4%) due to backbone differences. InternVL-2 shows smaller and inconsistent gains, improving from 14.4% (0.5B) to 21.3% (4B) but plateauing at 19.5% (20B), with the 4B model even surpassing the 7B in most reasoning types. Qwen2.5-VL exhibits weak scaling: performance increases modestly up to 32B (21.5%) but drops slightly at 72B (20.3%). Cross-family comparisons further reveal that larger size does not guarantee superiority, for example, Qwen2.5-VL-72B underperforms InternVL-2 4B.

We then analyze the impact of scaling the LLM backbone itself. As shown in Figure 6, backbone capacity correlates strongly with overall performance: larger backbones such as Qwen2.5-72B and 32B consistently achieve the highest scores, while smaller backbones (e.g., 0.5B, 1.5B) exhibit markedly weaker results. This positive correlation is particularly pronounced in *conceptual* ($\rho = 0.86$) and *hypothetical* ($\rho = 0.88$) reasoning, where scaling the backbone size almost monotonically improves performance across model series such as InternVL-3. In contrast, *quantitative* reasoning shows a weaker correlation ($\rho = 0.64$), indicating that simply enlarging the backbone does not translate into proportional gains and that quantitative reasoning likely requires stronger visual perception and advanced numerical reasoning beyond language capacity.

> **Takeaway: The Impact of Model Scaling**
>
> Larger models reliably boost conceptual/hypothetical reasoning, but quantitative gains remain weak and non-monotonic across series.

**The Impact of Modality.** To better understand how different input modalities contribute to scientific video reasoning, we conduct a series of ablation studies across text-only, audio-only, and video-only settings. For GPT-4o, the *option-only* condition yields an accuracy of only 10.5%, which is close to random guessing under our 10-option MCQA design and confirms that distractor construction does not leak information. The *visual-blind* setting achieves 15.8%, demonstrating that textual priors alone are insufficient. When provided with transcripts, GPT-4o attains 21.5%, outperforming visual-blind QA and indicating that narration adds helpful procedural cues. However, this remains far below the *video* condition, where GPT-4o reaches 24.9%, showing that visual evidence is the primary driver of performance. Providing the associated JoVE paper (*video+paper*) results in a slight improvement to 25.8%, suggesting that textual scientific descriptions offer marginal benefits for some conceptual or hypothetical questions, but the gain is minimal compared with the contribution of the video itself.

We further conduct modality ablations using Gemini-2.5-Pro to study the role of audio. The *audio-only* condition achieves 34.9%, substantially higher than transcript-only performance, reflecting that narrated experimental procedures provide more structured scientific information than raw text. Nevertheless, audio alone remains far below the *video-only* performance of 64.3%, confirming that visual observation of experimental setups, temporal dynamics, and measurement cues is indispensable. Combining both modalities (*audio+video*) achieves 67.0%, a modest gain over video alone, showing that audio augments visual reasoning but does not replace it.

Together, these ablations demonstrate that SciVideoBench questions are fundamentally *visual-dependent*. Textual or auditory signals can provide supportive procedural context, but accurate scientific reasoning requires grounding in the spatiotemporal evidence present directly in the video.

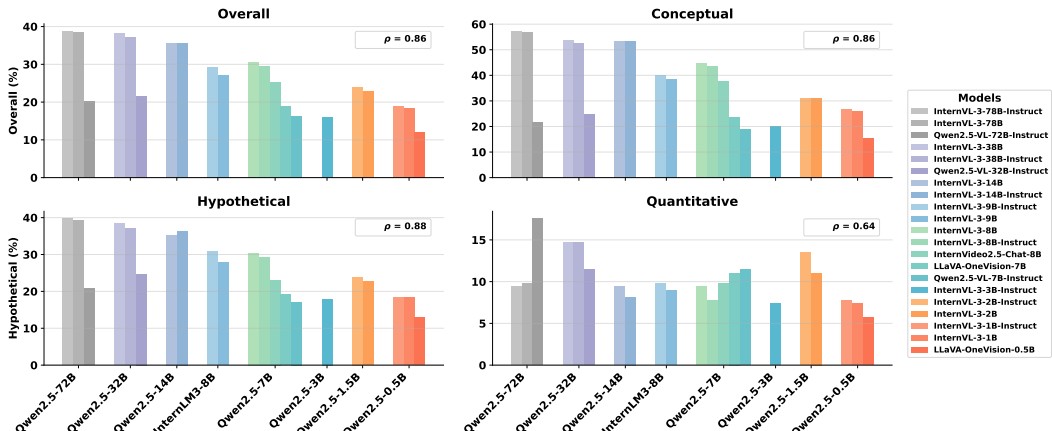

Figure 6: The impact of LLM backbones on the performance.

## 4.4 FAILURE CASE ANALYSIS

We illustrate a failure case with Gemini-2.0-Flash in Figure 7. The correct rationale is that the setup is a sealed desiccator containing a saturated potassium sulfate solution, which maintains a constant relative humidity of 98%; this environment is essential because the luminescent properties of the Ag-zeolite composite are highly hydration-sensitive, and cooling in this controlled atmosphere ensures reproducible results across experiments. The model fails despite partially recognizing the role of humidity control, ruling out "standardized high humidity". This error stems from two key failures: (1) ignoring on-screen evidence—the model overlooked the explicit "98% relative humidity" label

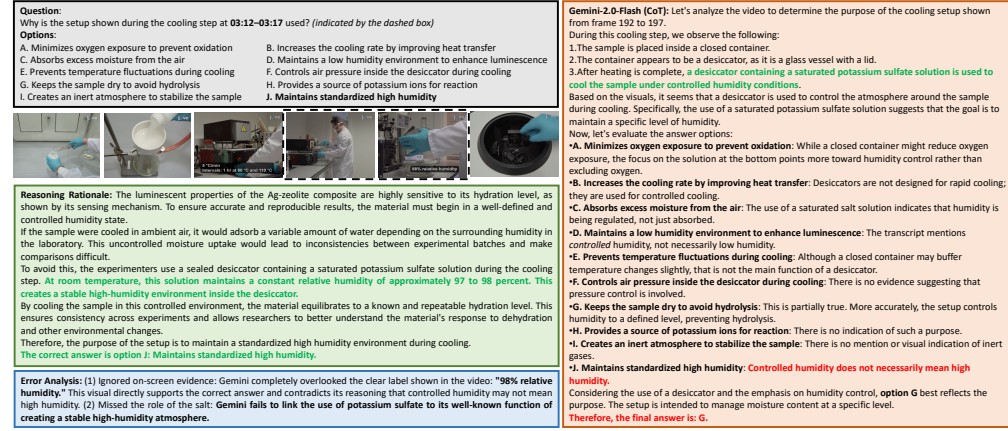

Figure 7: Failure case example of Gemini-2.0-Flash w/ chain-of-thought prompt.

in the video; and (2) misinterpreting the chemical function of potassium sulfate—generalizing it as generic moisture control rather than its well-established role in maintaining high humidity. This case highlights that solving such questions requires the integration of precise temporal localization (identifying the cooling step), fine-grained spatial perception (reading on-screen text), and expert-level domain knowledge (understanding the chemical's function), and demonstrates how current models often collapse when any one of these capabilities is lacking. More failure cases and the detailed failure pattern analysis can be found in Section F and K in the Appendix.

## 5 CONCLUSION

We present SCIVIDEOBENCH, the first scientific video reasoning benchmark that demands research-level knowledge to perform complex reasoning grounded in real-world experimental scenarios. SCIVIDEOBENCH bridges the gap between general-purpose video understanding and advanced scientific reasoning. It spans four major scientific disciplines and covers more than 25 distinct subjects, encompassing a broad spectrum of scientific inquiry. To enable rigorous evaluation, we design three complementary question types that reflect common reasoning challenges in scientific research. We evaluate a total of 35 models, including 5 proprietary and 30 open-source LMMs, to assess their current capabilities relative to expert human reasoning. Our results show that even the best model still struggles with these challenging tasks. We hope SCIVIDEOBENCH will serve as a milestone benchmark for advancing LMMs and inspire future research in the AI for Science community.

## 6 REPRODUCIBILITY STATEMENT

To support transparency and reproducibility, we document all key details of dataset construction and evaluation. The dataset includes full metadata for each video, including IDs, timestamps, annotations, and links to source content. All 1000 question–answer pairs are versioned and released with associated distractors and reasoning type labels. Evaluation scripts specify frame sampling strategies, input formatting, and decoding parameters (e.g., temperature, top-$k$) to ensure consistent reproduction of results. Experiments were conducted using up to $8\times$H100 GPUs, and we release prompts, seeds, and logs to minimize redundant computation and allow others to replicate our baselines. To maintain transparency, future updates will be versioned with detailed changelogs, and we will honor error corrections or takedown requests from rights holders.

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

## A  APPENDIX OVERVIEW

The appendix provides more details of our approach and additional experimental results, organized as follows:

## B  DISCIPLINES AND SUBJECTS

Table 3: List of subjects grouped by their corresponding disciplines.

| Discipline | Subjects |
|---|---|
| Biology | Biochemistry, Bioengineering, Biogeotechnology, Bioinformatics, Genetic Engineering, Lipidomics, Mechanobiology, Microfluidics, Mycology, Neurohistology, Neuroscience, Phycology, Proteomics, Regenerative Biology, Structural Biology, Cell Biology, Molecular Biology |
| Chemistry | Analytical Chemistry, Electrochemistry, Green Chemistry, Nanomaterials, Organic Chemistry, Photocatalysis, Photovoltaics, Physical Chemistry, Radiochemistry, Materials Chemistry |
| Medicine | Biomaterials, Cardiovascular Research, Dentistry, Drug Delivery, Immunology, Molecular Imaging, Nanomedicine, Oncology, Ophthalmology, Pharmacology, Radiopharmaceuticals, Regenerative Medicine, Surgical Devices, Therapeutics |
| Physics | Acoustofluidics, Additive Manufacturing, Aerosol Science, Applied Physics, Ceramic Engineering, Ceramics, Civil Engineering, Condensed Matter Physics, Ecological Engineering, Electronics, Fluid Dynamics, Materials Science, Mechanical Engineering, Microfluidics, Nanofabrication, Plasma Physics, Semiconductor, Sensors, Soft Robotics, Thermal Engineering |

# C  ANNOTATION DETAILS

To generate high-quality, research-level question-answer pairs, we develop a semi-automatic annotation pipeline, as shown in Figure 2, that integrates video, aligned transcripts, and paired research papers to provide rich scientific and procedural context. This setup enables accurate and visually grounded question generation while reducing manual workload through a multi-agent system.

To ensure quality and guide the semi-automatic annotation process, we first manually annotate a small set of exemplar cases. Four human experts (PhD students from biology, chemistry, medicine, and physics, respectively) review four selected papers (one for each subject) to extract key scientific concepts and procedural steps, then identify corresponding visual cues in the associated videos to construct sophisticated questions. These exemplar QA pairs are further refined and expanded via GPT-4o, resulting in 12 high-quality examples per question type. These are used as prompts to guide the human-in-the-loop multi-agent annotation system.

In the second stage, we assign distinct roles to a set of large language model agents, each responsible for a specific subtask in the annotation pipeline:

• **QA Generator** produces initial question–answer (QA) pairs from multimodal inputs—video, transcript, and associated paper text—guided by curated exemplar questions. Specifically, we design prompts for Gemini 2.5 Pro to first identify the core scientific question along with key aspects related to theory, experimental operations, and relevant numerical calculations for each video (the prompt is shown in Figure 8). Using these extracted elements and human-designed exemplars as references, Gemini 2.5 Pro then generates open-ended questions. To expand these into multi-choice format, the model is prompted to create nine scientifically plausible but incorrect distractors for conceptual and hypothetical reasoning (the prompt is shown in Figure 9), while for quantitative reasoning, distractors are automatically generated by adding appropriate Gaussian noise to the correct answer.

• **Evaluator** attempts to answer the generated question using the video, the transcript, and paper content, while also producing a detailed rationale that outlines the reasoning process. This step is to ensure all of the questions are answerable given sufficient information.

• **Visual Comparer** verifies whether the rationale is grounded in the video content by checking that all necessary visual cues are present. It also provides precise timestamps corresponding to each visual reference.

• **Refiner** refines the QA pairs by replacing generic or descriptive terms in the question with explicit references to visual segments, ensuring that the question cannot be answered without the video content. It also refines the options for each question to ensure the difficulty of the distractors. The prompt is shown in Figure 10.

• **Human Verifier** reviews the refined QA pairs and the outputs from the Visual Comparer to confirm that the question is fully grounded in the visual evidence. Specifically, these verifiers first check the timestamps annotation in questions to make sure the question is answerable based on the video content. If critical details (e.g., mass or temperature) are mentioned in the transcript but not visible in the video, the expert triggers a script to overlay this information directly onto the relevant video frames. Furthermore, the answer will be scrutinized to ensure that it matches the video content and the reasoning path.

In the final stage, the human experts perform a final review to manually correct any errors and improve clarity. The expert also ensures that each question is closely aligned with the core scientific contribution or focus of the experiment.

```
You are a domain expert in scientific video analysis. I will give you a scientific
experimental video, and your job is to perform a comprehensive multi-step analysis to
extract both procedural and conceptual insights.

Your tasks:

1. **Identify the Scientific Context:**
- Determine the **title** of the experiment.
- Identify the **discipline** (e.g., chemistry, biology, physics, etc.).
- Identify the **fine-grained subject** or subfield (e.g., analytical chemistry, cell
biology, fluid mechanics, etc.).
- Identify the core quantitative equation that can be used to analyze the data (e.g.,
if this is a chemical synthesis process, the chemical reaction formula is important,
such as C6H12O6 + 2H2O -> 6CO2 + 6H2O).
- The equations could be more than 1 in a video.

2. **Timestamp-Level Experimental Breakdown:**
- Break the video down into meaningful **temporal segments**.
- For each segment, describe what is happening in detail.
- Include **quantitative information** where possible (e.g., "00:03 - 00:14: 2.5 mL
of HCl is added to the beaker.").

3. **Key Experimental Points (3 Dimensions):**
- a. **Quantitative Aspects**: Describe measurable quantities used, such as volume,
time, temperature, concentrations, and any calculations that could be derived.
- b. **Operational Aspects**: Describe the procedures, tools, instruments, and
handling techniques shown in the experiment.
- c. **Scientific Principles**: Explain the underlying scientific concepts or
mechanisms behind the observed steps (e.g., acid-base neutralization, diffusion,
catalytic reaction, etc.).

Only return the JSON. Make sure all values are extracted and inferred from the video
with the highest possible accuracy.
```

Figure 8: The prompt for Gemini 2.5 Pro to generate metadata to support initial annotation.

```
You are an expert science educator.
Your task is to generate **10 multiple-choice options (A-J)** for the following question. Do **not** change the
question.
- Do not change the correct answer text.
- For the 9 distractors, you must create **scientifically plausible but incorrect** answers.
- Each distractor should reflect a distinct **misinterpretation**, such as:
  - misreading the video at a nearby but incorrect timestamp,
  - misunderstanding a visual cue,
  - applying an incorrect but believable scientific assumption.
  - The generated options should also be a short phrase as the correct answer.
Use the `timestamp_breakdown` below as your only context to design the distractors.
Return your answer in **valid JSON** format with:
- An `"options"` dictionary from "A" to "J"
- An `"answer"` field containing the **correct letter only** (e.g., `"C"`)
Example format:
{{
 "options": {{
   "A": "...",
   "B": "...",
   ...
   "J": "..."
  }},
 "answer": "C"
}}
Do not include any explanation or commentary.
---
Question:
{question_text}
Correct Answer Text:
{answer_text}
Why it is correct:
{reasoning_text}
```

Figure 9: The prompt for Gemini 2.5 Pro to generate distractors.

Listing 1: Example metadata generated from Gemini 2.5 Pro

```
1  {
2    "66497": {
3      "title": "Synthesis and Characterization of Self-Assembled
            Metal-Organic Framework Monolayers Using Polymer-Coated
            Particles",
4      "discipline": "Chemistry",
5      "subject": "Materials Chemistry",
6      "core_equation": "UiO-66-DDMAT + n(CH2=CHCOOCH3) --(
            Photocatalyst, Light)--> UiO-66-p(CH2-CH(COOCH3))n",
7      "timestamp_breakdown": [
8        "00:00 - 01:58: An introduction to the research...",
9        "01:59 - 02:05: 10 mg of catechol-DDMAT is weighed...",
10       "02:05 - 02:10: 5 mL of chloroform is added...",
11       "...",
12       "06:13 - 06:48: After the toluene evaporates, a freestanding
            monolayer forms..."
13     ]
14   }
15 }
```

You are given a scientific multiple-choice question that includes timestamp references to specific video moments (e.g., "(15:06)" or "4:30-4:50)").
Your task is to rewrite the question to make it **concise, abstracted, and timestamp-grounded**.
Specifically:
1. **Identify any descriptive phrases that explain or summarize what is shown at the given timestamps.**
2. **Replace those descriptive phrases with short, neutral references like:**
   - "the operation shown at [timestamp]"
   - "the behavior shown at [timestamp]"
   - "the outcome observed at [timestamp range]"
   - "the phenomenon illustrated at [timestamp]"
3. **Do not remove the timestamp.** The timestamp should remain to ensure grounding to the video.
4. Your goal is to:
   - Make the question unanswerable without viewing the video;
   - Avoid leaking any interpretative or explanatory information from the original wording;
   - Shorten and simplify the question for clarity.
Return the result as a JSON object:
{{
  "updated_question": "UPDATED_QUESTION_TEXT"
}}
---
### Example Input → Output
**Original:**
"What fundamental property of the circadian system is revealed by the difference between wild-type activity in the LD cycle (15:06) and the arrhythmic mutant (15:48)?"
**Rewritten:**
{{
  "updated_question": "What fundamental property of the circadian system is demonstrated by the differences shown at 15:06 and 15:48?"
}}
Question:
{question_text}

Figure 10: The prompt for Gemini 2.5 Pro to update distractors.

## D  DETAILED STATISTICS

We collected a total of 241 experimental videos spanning four major domains: Physics, Chemistry, Biology, and Medicine. These videos cover more than 25 distinct scientific subjects, as illustrated in Figure 3. The videos in SCIVIDEOBENCH have a competitive average duration of 484 seconds, with the duration distribution shown in Figure 11. This relatively long temporal scale ensures that the benchmark reflects the complexity and extended reasoning often required in real-world scientific experiments.

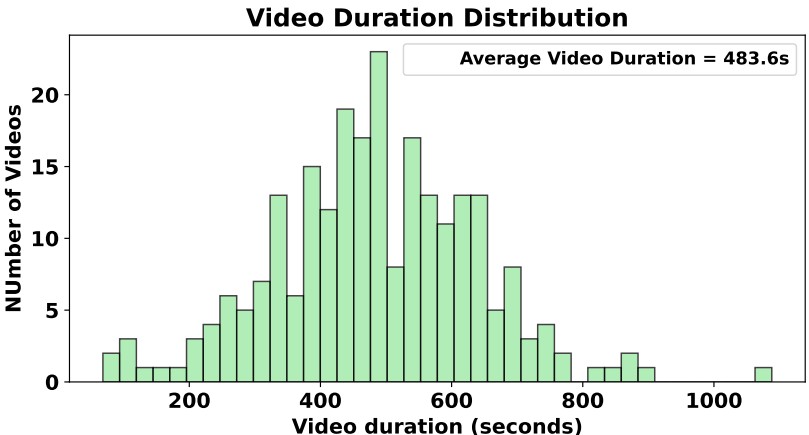

Figure 11: Video duration distribution in SCIVIDEOBENCH.

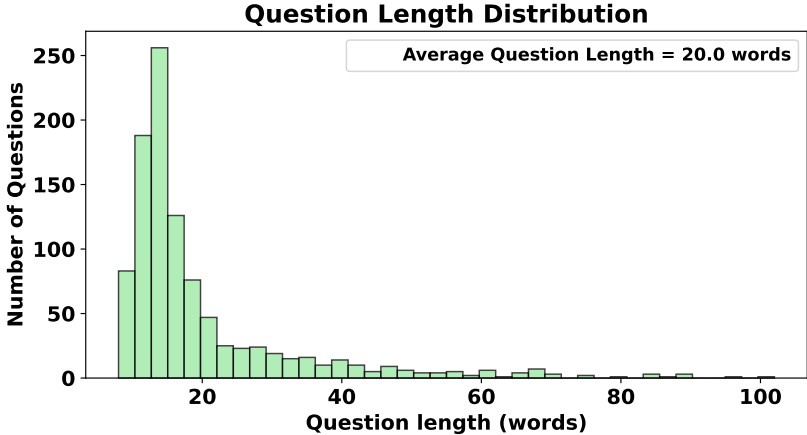

Figure 12: Question length distribution in SCIVIDEOBENCH.

Building upon these videos, we annotated a total of 1,000 challenging questions that demand research-level knowledge for both perception and reasoning. As shown in Figure 12, the average question length demonstrates that the questions are more linguistically complex than those in existing benchmarks, reinforcing the emphasis on detailed experimental understanding. Furthermore, the option length distribution in Figure 13 highlights the balanced design of ground-truth answers and distractors, with comparable average lengths, thereby reducing annotation bias and ensuring fair evaluation. To further capture the nature of academic research and experimental analysis, we carefully designed three distinct question types that reflect common reasoning scenarios observed across the videos, as illustrated in Figure 4. Collectively, these design choices ensure that the question–answer pairs in SCIVIDEOBENCH require deeper domain expertise and multi-step reasoning,

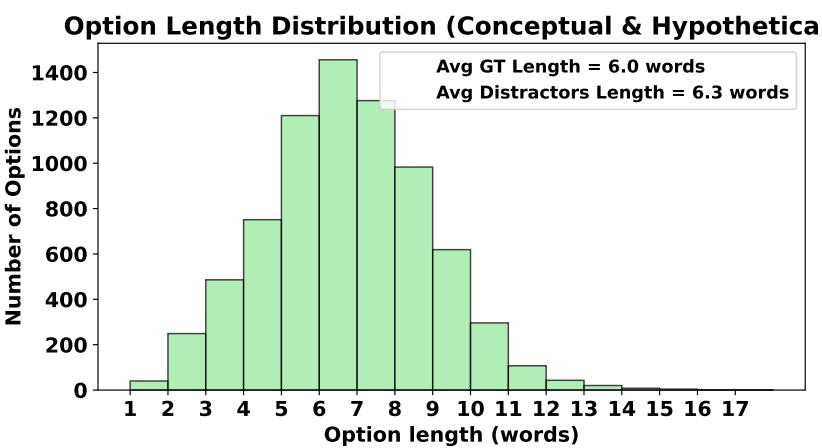

Figure 13: Option length distribution in SCIVIDEOBENCH.

distinguishing it from prior benchmarks that mainly target elementary or undergraduate-level understanding.

**245 Quantitative Reasoning** involves numeric perception, reasoning, and calculation. For a specific quantitative question, we require that all the numeric information must come from the video, which naturally ask the model to perceive informative values from the video before performing complicated calculations and reasoning.

**385 Hypothetical Reasoning** focuses on specific experimental operations that play a critical role for the motivation or outcome of the whole experiment. This question type usually involves hypothesized error, what-if analysis, or experimental control logic, which requires both expert-level knowledge on a specific domain and accurate visual perception to capture important details in the videos.

**370 Conceptual Reasoning** explore the mechanisms, protocols, and scientific principles behind the operations in the experiment that are inferred via visual/operational cues.

# E  CONFIGURATION OF EVALUATED MODELS

All of the evaluated models in this paper were released after May 2024. The detailed configuration of the evaluated models are shown in Table 4. We follow the default frame sampling settings of the official implementation of each model. The temperature is set to 0 to ensure stable predictions.

Table 4: Models used in our evaluation with confirmed release dates where available.

| Organization | Model | Release Date | Input Frames |
|---|---|---|---|
| *Proprietary Models* | | | |
| Google | Gemini-2.5-Pro | June 17, 2025 | 1fps |
| Google | Gemini-2.5-Flash | June 17, 2025 | 1fps |
| Google | Gemini-2.0-Flash | February 5, 2025 | 1fps |
| Google | Gemini-1.5-Pro | September 24, 2024 | 1fps |
| OpenAI | GPT-4o | May 13, 2024 | 256 |
| *Open-source Multimodal Models* | | | |
| OpenGVLab | InternVL-3 series | April 11, 2025 | 32 |
| Alibaba | Qwen2.5-VL | January 28, 2025 | 768 |
| OpenGVLab | InternVideo2.5 | January 21, 2025 | 512 |
| LMMS-Lab | LLaVA-OneVision | August, 2024 | 32 |
| OpenGVLab | InternVL-2 series | July 4, 2024 | 16 |
| LMMS-Lab | LongVA (7B) | June 24, 2024 | 128 |
| UCSD/CMU | LLaVA-NeXT-Video | May 10, 2024 | 32 |

## F   MORE FAILURE CASE STUDIES

In this section, we showcase more failure case among Conceptual, Hypothetical, and Quantitative Reasoning.

**Case 1: Misinterpretation of Spectral Evidence.**   InternVL-3-14B misread the electroluminescence spectrum by hallucinating multiple peaks and voltage-dependent shifts, concluding that multiple emissive species contributed to the emission. In reality, the spectrum displayed a single stable peak, indicating well-confined recombination. The model failed to recognize confinement as the key property and instead introduced instability, leading to the wrong answer. This reflects a tendency to over-generalize from prior knowledge of spectroscopy rather than grounding reasoning in the observed evidence.

**Case 2: Misunderstanding Experimental Procedure.**   Qwen2.5-VL-32B erred in interpreting the role of a 20-minute incubation step. It concluded that failure of incubation would prevent substrate hydrolysis, even though the substrate was not yet present during this period. The correct rationale emphasized pre-incubation of inhibitors with elastase, ensuring binding equilibrium before substrate addition. The model ignored this inhibitor-binding context, conflating incubation with general reaction progress, which resulted in selecting an irrelevant option.

**Case 3: Incorrect Solvent Accounting.**   InternVL-3-9B miscalculated the total solvent volume in a polymerization setup. It fabricated multiple additions of 1,4-dioxane and neglected the actual solvents present in the protocol (DMSO and DMF). Furthermore, it mishandled the inclusion of solvent-containing stock solutions and confused monomer additions with solvents. This misidentification led to both an incorrect summation and a mismatch between its numerical reasoning and the final selected option. The correct calculation, grounded in protocol details, yielded a total of 4.462 mL, which the model overlooked.

**Conclusion.**   Across these cases, a consistent pattern emerges: models often fail due to misalignment between the experimental context and the reasoning they generate. Errors stem from three main sources: (1) hallucination of experimental features not present in the input (spectral peaks, solvent additions), (2) failure to account for the temporal sequence of steps (substrate not present during incubation), and (3) confusion between categories of experimental components (monomers vs. solvents). These findings highlight the importance of grounding reasoning strictly in the provided experimental evidence rather than relying on generic domain priors. Improved training on step-by-step alignment between protocol details and answer derivation may reduce these systematic failures.

**Question**:
What does the property demonstrated by the electroluminescence spectrum at **06:38** imply about the charge recombination zone within the device? *(indicated by the dashed box)*
**Options**:
A. The recombination occurs predominantly at the interface layers
B. Charge recombination zone migrates causing broadening of emission spectrum
C. Charge recombination zone shifts with voltage causing multiple emissive states
D. Emission originates from degradation products varying with applied field
E. Electroluminescence derives mainly from exciplex states formed at interfaces
F. Charge recombination is well-confined and occurs from a single emissive species
G. Multiple emissive species contribute, indicated by fluctuating peak positions
H. Different host molecules emit at various voltages resulting in peak shifts
I. Unstable recombination leads to varying emission colors over time
J. Charge recombination is not confined and occurs throughout the device volume

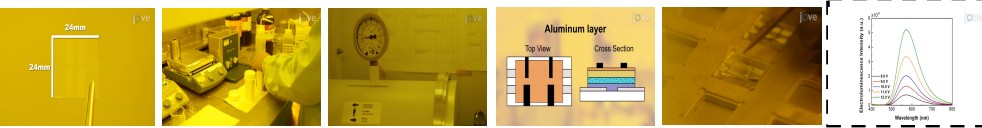

**InternVL-3-14B (CoT):** The electroluminescence spectrum shown in the video **demonstrates multiple peaks at different wavelengths**, which suggests that there are multiple emissive species within the device. This is because each peak corresponds to a different energy level of emission, which typically indicates different materials or states within the device that are emitting light. The shifting of these peaks with voltage further supports the idea that the charge recombination zone is not confined to a single area but rather involves multiple species that contribute to the overall emission.
Therefore, **the final answer is G**. Multiple emissive species contribute, indicated by fluctuating peak positions.

**Gemini-1.5-Pro (CoT):** The question asks about the implications of the electroluminescence spectrum shown at frame 394 for the charge recombination zone. This plot of emitted light as a function of wavelength for different voltages demonstrates that the wavelength of peak emission does not change. This suggests the device is optically stable. This means that regardless of the voltage applied, **the peak of the emitted light remains at the same wavelength**. A single, unchanging peak indicates a single emissive species and a well-confined recombination zone. **Therefore, the final answer is: F.**

**Reasoning Rationale:** The electroluminescence spectrum of the device provides insight into whether the charge recombination zone is stable or shifting under different applied voltages. If the recombination zone were moving or involving multiple emissive states, the emission spectrum would show shifts or multiple peaks corresponding to different sites such as host emission, exciplexes, or degradation products. This would compromise the stability of emission color across operating conditions. In the observed spectrum, however, **a single stable emission peak** is maintained across voltages. **This indicates that recombination is successfully confined to the DPTZ-DBTO2 guest molecules embedded in the CBP host matrix.** Such confinement prevents spectral drift, demonstrating that the device is well-architected to restrict recombination to one emissive species. This stability is particularly important for display applications, where consistent emission color at all brightness levels is essential for visual quality and device reliability. Therefore, the spectrum implies that the charge recombination is well-confined and occurs from a single emissive species.
**The correct answer is option F: Charge recombination is well-confined and occurs from a single emissive species.**

**Error Analysis:** (1) **Misinterpretation of spectral features:** InternVL-3-14B claimed that the electroluminescence spectrum shows multiple peaks and voltage-dependent shifts, concluding that multiple emissive species contribute. This directly contradicts the observed evidence described in the rationale, where a single stable peak is maintained across voltages. The model effectively hallucinated extra spectral complexity that was not present. (2) **Failure to recognize confinement as the key property:** Instead of identifying that stability of the emission peak demonstrates confinement of recombination to a single emissive species, the model assumed instability and distributed recombination.
This inverted the correct reasoning, leading to the wrong answer (G) instead of the correct answer (F).

Figure 14: Failure case example of InternVL-3-14B with chain-of-thought prompt compared with the correct reasoning of Gemini 1.5 Pro. InternVL-3-14B makes the wrong predictions because of the incorrect visual perception of the experiment results, while Gemini 1.5 Pro correctly captures the unchanged state of the peak at different wavelengths.

**Question**:
What could happen if the 20-minute incubation period described at 02:25 fails? *(indicated by the dashed box)*
**Options**:
A. Substrate and enzyme do not reach steady-state kinetics
B. Enzyme fails to degrade impurities in solution
C. Competitive binding between substrate and inhibitor does not occur
D. Enzyme is not activated by cofactors
E. Compounds do not bind to elastase
F. Enzyme does not partially denature before reaction starts
G. Reaction mixture temperature remains uneven
H. Substrate does not undergo initial hydrolysis
I. Plant extracts do not completely dissolve
J. Fluorescence equilibrium of the assay is not reached

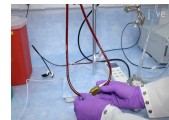 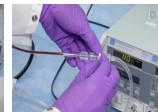 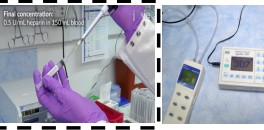 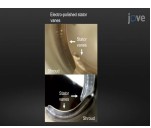 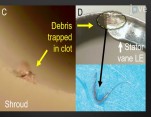

**Qwen2.5-VL-32B (CoT):** To answer this question, let's break it down step by step:\n\n### Step 1: Understand the Context\nThe question refers to a 20-minute incubation period described at frame 145. Incubation periods in biochemical or enzymatic assays are typically used to allow reactions to proceed under controlled conditions. This period might be crucial for achieving specific outcomes, such as reaching steady-state kinetics, ensuring complete dissolution of reagents, or allowing enzymes to interact with substrates.\n\n### Step 2: Analyze Each Option\nLet's evaluate each option to determine which one is most likely to occur if the 20-minute incubation period fails:\n\n#### A. **Substrate and enzyme do not reach steady-state kinetics**\n- Steady-state kinetics require enough time for the enzyme-substrate complex to form and for products to accumulate. If the incubation period is insufficient, the system may not reach steady state.\n- **Plausible** but depends on the specific context of the experiment.\n\n#### B. **Enzyme fails to degrade impurities in solution**\n- Some enzymes may degrade impurities during an incubation period. However, this is not a universal requirement for all experiments.\n- **Less relevant** unless the experiment explicitly involves impurity degradation.\n\n#### C. **Competitive binding between substrate and inhibitor does not occur**\n- Competitive inhibition requires the presence of both substrate and inhibitor. If the incubation period is too short, competitive binding might not occur fully, but this is highly dependent on the specific experimental setup.\n- **Plausible** but context-dependent.\n\n#### D. **Enzyme is not activated by cofactors**\n- Activation by cofactors often requires time for binding and activation. If the incubation period is insufficient, the enzyme might remain inactive.\n- **Plausible** if the enzyme requires cofactors.\n\n#### E. **Compounds do not bind to elastase**\n- Elastase is a specific enzyme that may require time to bind to its substrates or compounds. If the incubation period is too short, binding might not occur.\n- **Plausible** if the experiment involves elastase.\n\n#### F. **Enzyme does not partially denature before reaction starts**\n- Partial denaturation can sometimes be a necessary step for enzyme activity, but this is not a common requirement for most enzymes.\n- **Less relevant** unless the experiment specifically involves denaturation.\n\n#### G. **Reaction mixture temperature remains uneven**\n- Temperature control is important, but this is more about experimental setup than the incubation period itself.\n- **Less relevant** unless the incubation period is directly tied to temperature stabilization.\n\n#### H. **Substrate does not undergo initial hydrolysis**\n- Hydrolysis is a key step in many enzymatic reactions. If the incubation period is too short, the substrate might not undergo initial hydrolysis.\n- **Plausible** and directly related to the reaction progress.\n\n#### I. **Plant extracts do not completely dissolve**\n- Dissolution of plant extracts might require time. If the incubation period is insufficient, incomplete dissolution could occur.\n- **Plausible** if the experiment involves plant extracts.\n\n#### J. **Fluorescence equilibrium of the assay is not reached**\n- Many assays rely on fluorescence measurements. If the incubation period is too short, the fluorescence signal might not reach equilibrium.\n- **Plausible** and directly related to the assay outcome.\n\n### Step 3: Determine the Most Likely Outcome\nThe most general and universally applicable consequence of a failed incubation period is that the system might not reach the desired equilibrium or steady state. Among the options, **H. Substrate does not undergo initial hydrolysis** is a direct and fundamental issue that would arise if the incubation period is insufficient. Hydrolysis is a critical step in many enzymatic reactions, and without it, the reaction cannot proceed as intended.\n\n### Final Answer:\n$$\boxed{H}$$

**Reasoning Rationale:** The 20-minute incubation period is designed to **allow the elastase enzyme to interact with test compounds or the positive inhibitor PMSF before the substrate (SANA) is introduced**. Binding between an enzyme and its inhibitor is not always instantaneous; it is a dynamic process that requires time to reach equilibrium. This pre-incubation ensures that inhibitors occupy the enzyme's active site in advance, maximizing their inhibitory effect. If the incubation step fails or is skipped, the inhibitors will not have sufficient time to bind effectively. As soon as the substrate is added, it will compete directly with the inhibitor for access to the enzyme. This leads to a situation where inhibitory compounds cannot exert their true potency, and the assay results will underestimate their effect. The critical operational failure is that the compounds do not properly bind to elastase as intended. Therefore, if the 20-minute incubation fails, compounds do not bind to elastase. **The correct answer is option E: Compounds do not bind to elastase.**

**Error Analysis:** (1) **Misinterpretation of the incubation step**: Qwen2.5-VL-32B concluded that a failed incubation would prevent substrate hydrolysis (option H). This is incorrect because the substrate is not present during the incubation period, so hydrolysis cannot be affected at this stage. The model misunderstood the temporal sequence of the experiment.
(2) **Failure to recognize inhibitor pre-binding as the key purpose**: The rationale shows that the 20-minute incubation ensures inhibitors or test compounds bind elastase before substrate addition. The model ignored this pre-incubation role and instead generalized incubation as part of reaction progress.
This inverted the correct reasoning, leading to the wrong answer (H) instead of the correct answer (E).

Figure 15: Failure case example of Qwen2.5-VL-32B-Instruct. The model makes the wrong prediction because of lacking the domain knowledge about the incubation process.

**Question**:
What is the total volume of all liquid solvents added to the round bottom flask during the polymerization setup phase, prior to the degassing step?
**Options**:
A. 4.46 mL.   B. 6.15 mL.   C. 4.92 mL.   D. 5.91 mL.   E. 3.81 mL
F. 4.70 mL.   G. 2.91 mL.   H. 1.1 mL   I. 4.79 mL   J. 5.48 mL

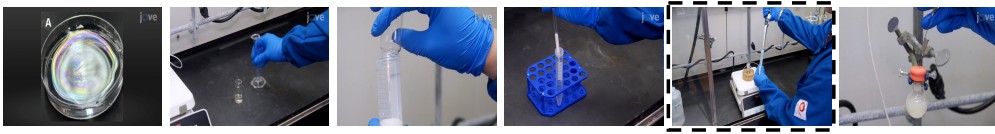

**InternVL-3-9B (CoT):** To determine the total volume of all liquid solvents added to the round bottom flask during the polymerization setup phase, we need to sum up the volumes of each solvent added. The video shows the addition of 1.1 mL of 1,4-dioxane, 1.1 mL of 1,4-dioxane, and 2.91 mL of 1,4-dioxane. **Adding these volumes together:\n\n1.1 mL + 1.1 mL + 2.91 mL = 5.11 mL\n\nHowever, the closest option to this sum is 5.48 mL. Therefore, the correct answer is D**

**Reasoning Rationale:** This question requires careful tracking of all solvent-containing components added during the polymerization setup. The video shows the following additions: 1. At 03:24, 2 mL of the UiO-66-DDMAT particle dispersion, which is stated to be in DMSO solvent. 2. **At 03:42, 12 microliters (0.012 mL) of the iridium catalyst stock solution is added**. The protocol clarifies this stock is in DMF. 3. At 03:49, 0.45 mL of the DDMAT stock solution is added, which is also in DMF. 4. At 04:02, 2 mL of DMSO is used to dissolve the methyl acrylate monomer. Methyl acrylate itself is the monomer, not a solvent. Therefore, the total volume of solvents is the sum of the volumes of DMSO and DMF added: 2 mL (from particle dispersion) + 0.012 mL (from catalyst stock) + 0.45 mL (from DDMAT stock) + 2 mL (for dissolving monomer) = 4.462 mL. **Thus, the answer is A.**

**Error Analysis:** (1) **Wrong solvent identification and omissions:** InternVL-3-9B hallucinated three additions of 1,4-dioxane (1.1, 1.1, 2.91 mL) and summed those, ignoring the actual solvent-bearing additions shown in the protocol (DMSO from the particle dispersion and monomer dissolution; DMF from the catalyst and DDMAT stocks). It also excluded the μL-scale stock addition and the dispersion solvent, both of which count "prior to degassing." (2) **Incorrect inclusion/exclusion logic:** The model treated bulk "1,4-dioxane" as the solvent of interest and implicitly ignored that the monomer is **not** a solvent, whereas the rationale correctly counts only solvent volumes (DMSO + DMF) and excludes methyl acrylate. (3) **Arithmetic/decision inconsistency:** After summing its invented volumes to 5.11 mL, the model claimed "the closest option is 5.48 mL" but then output **D (5.91 mL)**, revealing both numerical error and option mismatch.

Figure 16: Failure case example of InternVL-3-9B. The model makes the wrong predictions because of the wrong calculation logic and wrong solvent identification.

## G    RESULTS OF DIFFERENT QUESTION TYPES

As shown in the right portion of Table 2, Quantitative Reasoning consistently emerges as the most challenging category across models, regardless of architecture or scale. This difficulty is evident not only in proprietary models but also in open-source models, where Quantitative scores are systematically lower than those for Conceptual or Hypothetical reasoning. In fact, several open-source models perform close to or even below the random-guess baseline, for example, InternVL2-4B achieves only 9.39%, and InternVL-3-1B-Instruct reaches 7.76% in Quantitative Reasoning. By contrast, Conceptual Reasoning generally yields the highest scores, slightly ahead of Hypothetical Reasoning, although the gap is modest. These patterns indicate that SCIVIDEOBENCH places particularly high demands on precise numerical reasoning and multi-step calculation abilities, while Conceptual and Hypothetical questions—though still challenging—tend to be more approachable for current multimodal LLMs.

## H    RESULTS OF DIFFERENT DISCIPLINES

Across different disciplines, the performance trends remain relatively consistent, indicating that no single discipline presents a disproportionately higher level of difficulty in SCIVIDEOBENCH. For Gemini-2.5-Pro, Chemistry emerges as the lowest-performing discipline, whereas Medicine achieves the highest accuracy; while Gemini-1.5-Pro displays the opposite pattern, with its best results in Chemistry and comparatively lower performance in other disciplines. This contrast suggests that the impact of model architecture and scale can vary across domains, and that strengths in one discipline do not necessarily translate directly to others.

## I    THE IMPACT OF AUDIO

To investigate how audio, which provides additional information about the experiments shown in the video, will affect the model performance, we evaluate Gemini-2.5-Pro (Google, 2025) and Qwen2.5-Omni-7B (Jin Xu, 2025) due to their capability of supporting full modality input. From Table 5, we can observe consistent improvement brought by audio input for both Gemini-2.5-Pro and Qwen2.5-Omni-7B in overall performance (2.70% and 2.80%, respectively). This limited improvement also demonstrates that the visual content dominates the model performance.

Table 5: Model Performance Improvement by Audio on SCIVIDEOBENCH.

| Models | Use Audio | Overall | Question Type | | | Discipline | | | |
|---|---|---|---|---|---|---|---|---|---|
| | | | Conceptual | Hypothetical | Quantitative | Biology | Chemistry | Medicine | Physics |
| Gemini-2.5-Pro (Google, 2025) | N | 64.30 | 69.73 | 67.79 | 50.61 | 64.79 | 61.82 | 74.77 | 61.44 |
| Gemini-2.5-Pro (Google, 2025) | Y | 67.00 | 74.32 | 67.53 | 55.10 | 68.70 | 61.21 | 71.96 | 66.14 |
| Qwen2.5-Omni-7B (Jin Xu, 2025) | N | 14.70 | 16.69 | 14.97 | 9.74 | 14.39 | 14.36 | 17.85 | 14.25 |
| Qwen2.5-Omni-7B (Jin Xu, 2025) | Y | 17.50 | 20.29 | 18.51 | 12.80 | 18.24 | 16.52 | 18.19 | 17.67 |

## J  DISAGREEMENT BETWEEN CoT AND DIRECT ANSWER

To further investigate the possible reason that CoT prompting has a such different impact for different reasoning type for open-source models, we analyze the predictions of three representative models: InternVL-3-78B, InternVL2-Llama3-76B, and Qwen2.5-VL-32B-Instruct. From Table 6, we observe that chain-of-thought (CoT) prompting does not consistently improve evaluation performance across reasoning types. Globally, the number of instances where the direct strategy is correct but CoT is wrong slightly exceeds the reverse, indicating that CoT often introduces additional errors. A more detailed breakdown by reasoning type reveals that the negative impact is concentrated in *Conceptual* and *Hypothetical* reasoning. In these categories, CoT tends to amplify hallucinations or over-elaborate on causal relations, leading the model away from the correct choice, whereas direct answering can rely more on memorized factual knowledge or surface-level pattern recognition. In contrast, *Quantitative* reasoning benefits substantially from CoT: across the three models, we see more cases where CoT corrects errors that the direct approach would miss. This suggests that explicit reasoning chains help models externalize intermediate computational steps (e.g., arithmetic or logical comparisons), which are otherwise challenging when only a single final answer is required. A plausible reason why CoT hurts open-source models on conceptual and hypothetical reasoning is that open-source models are less robust in producing faithful long-form causal explanations, often hallucinating or over-elaborating when reasoning about abstract scenarios, but CoT provides a clear scaffold for step-by-step arithmetic or numerical comparisons, which directly reduces errors in quantitative tasks.

Table 6: Comparison of agreement and disagreement between CoT and Direct answers across three models.

| Model | Both Correct | Both Wrong Agree | Both Wrong Disagree | CoT Correct, Direct Wrong | Direct Correct, CoT Wrong |
|---|---|---|---|---|---|
| **Overall** | | | | | |
| InternVL-3-78B | 259 | 253 | 242 | 120 | 126 |
| InternVL2-Llama3-76B | 139 | 287 | 340 | 110 | 124 |
| Qwen2.5-VL-32B-Instruct | 96 | 335 | 338 | 112 | 119 |
| **Conceptual Reasoning** | | | | | |
| InternVL-3-78B | 149 | 77 | 41 | 42 | 61 |
| InternVL2-Llama3-76B | 58 | 125 | 97 | 43 | 47 |
| Qwen2.5-VL-32B-Instruct | 37 | 147 | 92 | 39 | 55 |
| **Hypothetical Reasoning** | | | | | |
| InternVL-3-78B | 93 | 102 | 88 | 44 | 58 |
| InternVL2-Llama3-76B | 48 | 114 | 129 | 38 | 56 |
| Qwen2.5-VL-32B-Instruct | 49 | 145 | 106 | 39 | 46 |
| **Quantitative Reasoning** | | | | | |
| InternVL-3-78B | 17 | 74 | 113 | 34 | 7 |
| InternVL2-Llama3-76B | 33 | 48 | 114 | 29 | 21 |
| Qwen2.5-VL-32B-Instruct | 10 | 43 | 140 | 34 | 18 |

## K ERROR ANALYSIS

To gain an in-depth understanding of how far the current models are from real human experts, we comprehensively analyze the results of Gemini-1.5-Pro-Thinking and Gemini-2.0-Flash-Thinking. Comparing these reasoning steps with the real rationale from the annotation process, which was carefully verified by human experts, we found three noticeable errors: Incorrect Visual Perception, Inaccurate Reasoning, and Lack of Domain Knowledge. Importantly, most of the wrong responses stem from a complex error combination, e.g., both incorrect visual perception and inaccurate reasoning progress.

**Incorrect Visual Perception (70.68%)** is the most common factor that leads to an incorrect conclusion. It leads to misinterpret of what is visually shown in the video and identify the wrong moment in the temporal span. For instance, in Figure 7, Gemini-2.0-Flash overlooks the on-screen textual information indicating the humidity, which leads to the consequence that it obtains incorrect option analysis for option J, maintaining standardized high humidity.

**Inaccurate Reasoning Progress (63.25%)** also constitutes a main portion of the error cause. It usually involves logical flaws despite correct or partially correct observations. In Figure 7, Gemini-2.0-Flash fails to establish the connection between the operation of using potassium sulfate and the high-humidity consequence.

**Lack of Domain Knowledge (49.40%)** happens when models fail to connect the visual evidence with specific expert knowledge that helps answer the question. For example, in Figure 7, the function of potassium sulfate should be a well-known expert knowledge for researchers; however, Gemini-2.0-Flash is not able to analyze even perceive the existence of the sulfate in the video, which is strong evidence of domain knowledge lacking.

## L  COPYRIGHT

All videos used in SCIVIDEOBENCH originate from the Journal of Visualized Experiments (JoVE), which retains the copyright to the original recordings. Our benchmark leverages these videos strictly under fair use for non-commercial research and educational purposes, in line with academic standards. The dataset does not redistribute full videos; instead, it provides structured annotations, metadata, timestamps, and video IDs that allow authorized users to access the original JoVE content via institutional or personal subscription. Where minimal frame thumbnails are necessary for unambiguous visual grounding, only the smallest subset of frames is included. This design respects copyright while still enabling reproducible scientific research. We emphasize that SCIVIDEOBENCH is strictly intended for research and educational use, and must not be used for commercial applications.

## M  ETHICAL CONSIDERATIONS

SCIVIDEOBENCH is carefully curated to avoid sensitive or personally identifiable content. The videos are professionally produced laboratory demonstrations published with editorial oversight, focusing on experimental procedures, equipment, and scientific phenomena rather than human subjects. While some clips may include researchers' hands or partial presence in the experimental setting, no personal, medical, or biometric data is exposed. We do not annotate or distribute personally identifying information, and any textual overlays included in the benchmark are limited to scientific variables (e.g., masses, temperatures) extracted from the accompanying manuscripts and transcripts to aid verification. Annotations are produced through a human-in-the-loop, multi-agent pipeline in which LLMs propose items, automated agents check answerability and visual grounding with timestamps, and human experts refine and verify the outputs. This design ensures data integrity, reduces hallucination, and enforces grounding in authentic scientific content. To mitigate dual-use risks, all questions are framed for reasoning and understanding rather than step-by-step procedural replication; the benchmark is not intended to serve as laboratory training material or to guide hazardous experimentation. Overall, the dataset adheres to responsible practices for scientific data collection and annotation, in alignment with JoVE's licensing policies and ethical guidelines.

## N  LIMITATION

Despite its contributions, SCIVIDEOBENCH has several limitations. First, its coverage reflects the domain distribution of JoVE and primarily English-language scientific materials, which may underrepresent other subfields, experimental practices, or linguistic contexts. Second, although our multi-agent, human-in-the-loop annotation pipeline was designed to ensure rigor, errors and biases may persist in question phrasing, answer distractors, or visual grounding. Third, the benchmark focuses on research-level laboratory scenarios and may not fully capture the breadth of scientific reasoning tasks in other modalities (e.g., field studies, computational simulations). We highlight these limitations to provide transparency, avoid overstating current capabilities, and encourage future work that broadens scientific coverage, reduces bias, and develops more efficient evaluation protocols.

## O  USE OF LLMS

In this work, large language models (LLMs) are employed exclusively within our semi-automatic annotation pipeline and for polishing the manuscript text. Specifically, LLMs assist in generating candidate question–answer pairs, producing rationales, and refining phrasing during annotation, always under human verification and correction. Beyond these annotation and writing-support roles, no additional use of LLMs is involved in the development of SCIVIDEOBENCH.

