# OpenReview forum: "SciVideoBench: Benchmarking Scientific Video Reasoning in Large Multimodal Models"
_ICLR.cc/2026/Conference — ICLR 2026 Conference Desk Rejected Submission_

### Official Review · Reviewer_1k1D · 2025-10-29

**Soundness:** 3
**Presentation:** 2
**Contribution:** 2
**Rating:** 4
**Confidence:** 4

**Summary:**

The paper introduces SciVideoBench, a benchmark targeting research-level scientific video reasoning. It comprises 1,000 multiple-choice questions drawn from research-grade experimental videos (sourced from JoVE) spanning physics, chemistry, biology, and medicine, with questions verified via a human-in-the-loop, multi-agent pipeline that enforces visual grounding with timestamps. The benchmark emphasizes conceptual, hypothetical, and quantitative reasoning beyond recognition. Reported results show: (i) very low human graduate-student accuracy (17.4%), (ii) “vision-blind” text-only baselines performing near chance (e.g., GPT-4o ~15.8%), (iii) a large gap between proprietary and open-source LMMs (e.g., Gemini-2.5-Pro 64.3% vs. best open 38.8%), and (iv) sizeable gains from chain-of-thought prompting for some models. The dataset redistributes annotations/metadata/timestamps (not full videos) under JoVE’s copyright constraints.

**Strengths:**

Originality. Focuses on scientific video reasoning with research-grade lab videos (tri-modal alignment: video, audio narration, and peer-reviewed manuscripts), addressing a gap in existing general video QA benchmarks.
Quality. The human-in-the-loop multi-agent annotation pipeline aims to ensure answerability and visual grounding (timestamps + visual comparer + expert verification).
Clarity. Scope and contribution are clearly stated; the paper documents evaluation settings and analyses (blind baselines, proprietary vs. open-source, CoT effects).
Significance. Results reveal substantial headroom (e.g., human grads at 17.4%; big proprietary–open gaps), underscoring the difficulty and potential impact for LMM research on scientific reasoning.

**Weaknesses:**

Scale & format. The benchmark is limited to 1k MCQs. At this difficulty level, multiple-choice can obscure partial competence and invite guessability. Consider expanding the scale and adding open-ended rationales or structured reasoning targets (e.g., numeric derivations, unit-checked answers) to better capture scientific reasoning depth.
Domain coverage constraints. Dependence on JoVE (English) narrows modality and topical breadth, which the authors also note as a limitation. Broader sources (field studies, simulations, non-English material) would reduce domain bias and improve generality.  ￼
LLM-in-the-loop artifacts. LLMs are used in annotation (generation, rationales, phrasing). While expert verification is present, quantify and stress-test for annotation artifacts (e.g., distractor style regularities) and measure inter-annotator agreement to ensure reliability beyond model-authored templates.
Text-only solvability & audio confounds. The blind baseline helps, but add stronger text-only and audio-only ablations per question type, plus adversarial filtering to ensure solutions truly require video evidence rather than transcripts or narration alone. (Appendix mentions “Impact of Audio” — consider promoting those analyses.)
Human evaluation design. The 17.4% graduate-student accuracy suggests extreme difficulty. Provide more detail: subject expertise matching, time limits, interface, allowed replay, and report per-domain human scores and inter-rater stats to calibrate realism vs. trickiness.

**Questions:**

Answerability audits. What fraction of items remain solvable from transcripts alone or audio alone? Can you release per-item tags indicating which modality is necessary/sufficient?
Annotation reliability. Please report inter-annotator agreement, human edit rates on LLM-proposed items, and an artifact probe (e.g., train a small text-only classifier on distractor style to test leakage).
Human study details. How were graduate students recruited and matched to subjects? What were the instructions/time budgets, and could they scrub video timelines? Any per-discipline human results?
Audio impact. The appendix lists “Impact of Audio.” Can you quantify modality contributions (video vs. audio vs. both) per reasoning category in the main text?

---

> ### Author Response · Authors · 2025-11-23
> **[1/5] Response to Reviewer 1k1D**
>
> We thank Reviewer 1k1D for the detailed and constructive evaluation of our work. Below we provide point-by-point responses to each weakness and question.
>
>
> ## **1. Benchmark scale & multiple-choice format**
>
> **Response:**
> We appreciate the reviewer’s concern. Our design choices follow established practices in recent high-quality video reasoning benchmarks and are aligned with both **feasibility** and **evaluation reliability**.
>
> #### **Benchmark scale (1k MCQs) is reasonable and comparable to prior work**
> Although 1,000 questions may seem modest, it is consistent with (or larger than) recent research-level video benchmarks. For example, **Video-MMMU** contains **900** high-difficulty questions across professional domains. Expanding SciVideoBench significantly beyond 1,000 MCQs is non-trivial, and each question requires detailed timestamp alignment, scientific validation, and multi-stage human verification, which imposes substantial annotation cost and expertise requirements.
>
> #### **Why multiple-choice QA (MCQA) instead of open-ended QA?**
> The field increasingly recognizes fundamental evaluation challenges with open-ended QA for complex multimodal tasks:
>
> - **Open-ended answers introduce high variability**, making evaluation error-prone and often unreliable. As highlighted by Zhang et al. [2], “accurate evaluation is difficult due to the variability in natural language responses.”
> - **Rule-based or lexical matching metrics fail** when answers involve multi-step reasoning, units, approximations, or procedural descriptions.
> - **LLM-as-a-judge evaluation**, while more flexible, introduces new biases, requires additional computational overhead, and can be unstable or inconsistent across models [3–5].
>
> Thus, MCQA remains the **most stable, scalable, and reproducible** paradigm for assessing high-level reasoning in video tasks.
>
> #### **Why 10-option MCQ? Reduced guessability and stronger discrimination**
> To mitigate the common concern that MCQ formats allow guessing, we use a **10-option format**, which reduces random-guess accuracy to **10%**, which is far below the ~25% seen in typical 4-choice benchmarks. Our **option-generation and distractor refinement pipeline** further ensures:
>
> - distractors are scientifically plausible
> - no answer-pattern leakage
> - no stylistic artifacts from LLM proposals
> - option-only accuracy stays near random (empirically ≈10%)
>
> This results in a *high-discriminative* evaluation suitable for research-level tasks.
>
> #### **Open-ended reasoning traces are available**
> Although the benchmark adopts MCQA for evaluation stability, SciVideoBench **already includes detailed reasoning chains** generated and human-verified during annotation. These will be released to enable future research on:
>
> - rationale evaluation
> - explanation-assisted QA
> - structured scientific reasoning
>
>
> ### **References**
>
> [1] Hu, K., Wu, P., Pu, F., et al. *Video-MMMU: Evaluating Knowledge Acquisition from Multi-Discipline Professional Videos.* arXiv:2501.13826, 2025.
> [2] Zhang, Y., Su, Y., Liu, Y., et al. *Automated generation of challenging multiple-choice questions for vision language model evaluation.* CVPR 2025.
> [3] Chen, D., Chen, R., Zhang, S., et al. *MLLM-as-a-Judge: Assessing multimodal LLM-as-a-judge with vision-language benchmark.* ICML 2024.
> [4] Laurençon, H., Marafioti, A., Sanh, V., et al. *Building and better understanding vision-language models: insights and future directions.* arXiv:2408.12637, 2024.
> [5] Mañas, O., Krojer, B., Agrawal, A. *Improving automatic VQA evaluation using large language models.* AAAI 2024.

---

> ### Author Response · Authors · 2025-11-23
> **[2/5] Response to Reviewer 1k1D**
>
> ## **2. Domain coverage constraints**
>
> **Response:**
> JoVE itself provides **substantial international diversity**. It publishes videos from **over 1,000 universities and research institutes worldwide**, and the subset included in SciVideoBench contains videos originating from **more than 12 countries**, including the United States, China, Japan, Australia, and the United Kingdom. This broad geographical representation ensures that the benchmark captures a wide range of experimental practices, equipment setups, and instructional styles across global research communities.
>
> Moreover, English serves as the **common working language of modern scientific communication**, particularly in STEM fields. Using English-language research videos aligns SciVideoBench with the dominant mode of dissemination in the scientific community and avoids confounding linguistic variability with the intended evaluation of multimodal scientific reasoning.
>
> Finally, SciVideoBench is intentionally centered on **real-world experimental scenarios**, reflecting authentic laboratory procedures and hands-on scientific processes. As such, *simulations, synthetic environments, and purely theoretical content are outside the current scope* of the benchmark. Our focus is to evaluate models on grounded, visually rich, experimentally anchored reasoning tasks, which is an area where existing benchmarks remain extremely limited.
>
>
> ## **3. LLM-in-the-loop artifacts & annotation reliability**
>
> **Response:**
> We appreciate the reviewer’s concern and clarify that SciVideoBench was designed with multiple safeguards to prevent LLM-induced artifacts and ensure annotation reliability. Importantly, **no question or distractor in the final benchmark is directly adopted from an LLM without human verification and revision**. Instead, Gemini-2.5-Pro serves only as a *draft generator*, and all content undergoes extensive human filtering and correction.
>
> #### **(1) Multi-stage filtering ensures LLM artifacts do not persist**
> Our pipeline initially produced **over 4,000 candidate Q&A pairs**, but only **1,000** survived the multi-stage verification process:
>
> - **Visual-blind filtering (GPT-4o):**
>   Candidate questions were tested without video input. If GPT-4o could answer correctly using only text, the question was rejected or revised.
>
> - **Option-only guess suppression:**
>   Early drafts exhibited high option-only solvability because the correct answer tended to be longer or more detailed.
>   To mitigate this:
>   - We used LLMs to generate *stronger distractors* that mirror the linguistic and scientific style of the correct answer.
>   - Human annotators iteratively revised distractors until **option-only accuracy dropped to near random (~10%)**, ensuring no leakage through pattern shortcuts.
>
> These steps effectively eliminate LLM-generated stylistic bias or predictable artifacts in the distractor set.
>
> #### **(2) Human-in-the-loop verification ensures correctness, grounding, and consistency**
> Five annotators with PhD-level STEM backgrounds validated every item. Verification focused on two core principles:
>
> - **Answerability:**
>   Ensuring each question can be answered **solely from the video evidence**, without relying on external scientific knowledge.
>
> - **Temporal grounding:**
>   Checking that all referenced actions, motions, operations, or measurements appear **precisely within the provided timestamps**. Any hallucinated or misaligned temporal reference triggered revision or rejection.
>
> #### **(3) Cross-validation provides practical inter-annotator agreement**
> While we did not report a numerical agreement score, the annotation workflow incorporates a **systematic cross-verification mechanism**:
>
> - Every batch is reviewed by a **second annotator**.
> - Any disagreement prompts a **structured resolution** involving an expert reviewer.
> - This multi-pass process acts as a strong practical surrogate for explicit inter-annotator statistics.
>
> Together, these mechanisms ensure that SciVideoBench’s annotations are **reliable, visually grounded, and free from LLM-induced artifacts**, despite the use of an LLM in early draft stages.

---

> > ### Author Response · Authors · 2025-11-23
> > **[3/5] Response to Reviewer 1k1D**
> >
> > ## **4. Text-only solvability & audio confounds**
> >
> > **Response:**
> > To address the reviewer’s concern on whether SciVideoBench questions can be solved without video evidence, we conducted additional modality-specific ablations using **GPT-4o** across five settings: **option-only**, **visual-blind**, **transcript-only**, **video**, and **video+paper** (results for the latter are shown in Section 4.3 of the main rebuttal). These experiments aim to dissect the contribution of textual, auditory, and visual modalities, and to test whether any shortcut solutions exist.
> >
> > The updated findings demonstrate:
> >
> > - **Option-only Guess accuracy is close to random (≈10%)**, confirming that the distractors do not leak textual patterns or stylistic cues.
> > - **Transcript-only QA significantly outperforms visual-blind**, showing that narration provides helpful scientific descriptions, but is still *insufficient* to answer most questions.
> > - **Quantitative questions show consistent accuracy across all non-visual settings (~11.8%)**, indicating that numerical values cannot be inferred without visually inspecting the experiment.
> >
> > Together, these observations reinforce that SciVideoBench questions are **not solvable through text, narration, or linguistic priors alone**, and that **visual, experiment-grounded reasoning is indispensable**.
> >
> > | Mode            | Overall | Conceptual | Hypothetical | Quantitative |
> > |-----------------|---------|------------|--------------|--------------|
> > | Option-only Guess | 10.50 | 11.35 | 10.65 | 8.98 |
> > | Visual-blind QA   | 15.80 | 14.05 | 20.00 | **11.84** |
> > | Transcript-only QA| 21.50 | 26.22 | 23.12 | **11.84** |
> > | Video QA          | 24.90 | 30.27 | 28.05 | **11.84** |
> >
> > These results collectively show that **SciVideoBench rigorously enforces visual dependence**, and that neither textual shortcuts nor transcript cues are sufficient to solve its research-level scientific questions.

---

> > > ### Author Response · Authors · 2025-11-23
> > > **[4/5] Response to Reviewer 1k1D**
> > >
> > > ## **5. Human study design: low graduate-student performance (17.4%)**
> > >
> > > **Response:**
> > > To ensure a fair and realistic human baseline, we recruited **three graduate students per discipline** (Biology, Chemistry, Physics, and Medicine). All participants were actively enrolled in STEM graduate programs and had general scientific training, but importantly, they were **not specialists in every sub-domain** represented in SciVideoBench. This reflects the intended evaluation setting: measuring the performance of *well-trained but non-specialist graduate students*, rather than domain experts for every specialized research area.
> > >
> > > **Instructions & evaluation setup:**
> > > - Participants were required to answer questions in a **strictly closed-book** setting.
> > > - They were given **only the videos, questions, and answer options**, with no access to JoVE papers, textbooks, notes, or online resources.
> > > - They were told they could **rewatch, scrub, and replay** each video freely, as many times as needed.
> > > - They completed all questions **within a single day**, using only standard video players without any custom tools or interfaces.
> > >
> > > This setup ensures that the human baseline reflects **pure video-based scientific reasoning**, not external knowledge lookup or text-based comprehension.
> > >
> > > **Why is the human accuracy (17.4%) low?**
> > > SciVideoBench intentionally targets **research-level experimental reasoning**, involving specialized procedures, precise temporal cues, implicit scientific assumptions, and quantitative analysis. The dataset spans **over 25 subjects**, many with deeply specialized subfields.
> > >
> > > For example, even **within a single discipline**, the level of specialization required by certain questions can exceed the background of a typical graduate student:
> > >
> > > - A graduate student specializing in **organic chemistry** may struggle with questions rooted in **analytical chemistry**, **physical chemistry**, or **chemical biology**, which require different conceptual and experimental expertise.
> > > - A **physics** student working on condensed matter may not be familiar with **optical instrumentation**, **fluid dynamics**, or **medical imaging physics** that appear in other videos.
> > > - A **biology** student focused on molecular genetics may not be able to answer questions involving **neuroscience assays**, **cell-imaging protocols**, or **biophysical measurement techniques**.
> > >
> > > Thus, even though students only answer questions within their own discipline, the **breadth and depth of sub-domains** in SciVideoBench inevitably introduce challenges that exceed the training of non-specialist graduate students.
> > >
> > > Given this diversity, it is **not feasible** to assemble a human cohort whose expertise uniformly covers every specialized research topic represented in the benchmark.
> > >
> > > Thus, the low human score does **not** indicate poor question calibration; rather, it highlights that:
> > > 1. **The benchmark assesses advanced, domain-specific, multimodal reasoning**, not basic scientific knowledge.
> > > 2. Even strong graduate students struggle when confronted with **research-level procedures outside their own specialization**, especially in a closed-book, video-only setting.
> > > 3. This difficulty level is appropriate for a benchmark designed to challenge frontier multimodal models.
> > >
> > > In summary, the human study setup was carefully designed to be fair, realistic, and tightly controlled. The low human accuracy is an expected outcome given the **research-grade scientific nature** of the tasks, and it underscores the value and difficulty of SciVideoBench.

---

> ### Author Response · Authors · 2025-11-23
> **[5/5] Response to Reviewer 1k1D**
>
> ## **6. Responses to Reviewer’s Questions**
>
> ### **Q1. Answerability audits**
> *“What fraction of items remain solvable from transcripts alone or audio alone? Can you release per-item modality tags?”*
>
> **Response:**
> Thank you for raising this important point. As discussed in our modality ablations, we evaluated **GPT-4o** under the *transcript-only* setting, which provides a useful **reference point** for how much information can be extracted purely from narration. While transcript-only accuracy is higher than visual-blind QA, it remains **substantially lower** than video-based performance, confirming that narration alone is not sufficient for most questions.
>
> However, transcript-only model performance **cannot be directly interpreted as a solvability label**, because:
> - Different models vary greatly in language understanding and narrative comprehension.
> - Transcript phrasing differs across JoVE videos, affecting different models differently.
> - The true solvability of a question without video evidence must be **human-verified**, not inferred from model output.
>
> To address this more systematically, we plan to release **per-item modality tags** in the next version of SciVideoBench. These tags will be derived through **explicit human verification**, indicating whether a question is **transcript-sufficient** or not.
>
> This will allow future users to distinguish modality dependencies more clearly and perform more fine-grained analyses.
>
>
> ### **Q2. Annotation reliability**
> *“Please report inter-annotator agreement, human edit rates on LLM-proposed items, and an artifact probe.”*
>
> **Response:**
> Please see the response to weakness 1.3.
>
>
> ### **Q3. Human study details**
> *“How were graduate students recruited and matched? What instructions, time budgets, replay ability? Per-discipline human scores?”*
>
> **Response:**
> Please see the response to weakness 1.5 for the details of human evaluation. For the per-disciline human scores, we show the accuracy for each discipline in the main table.
>
>
> ### **Q4. Audio impact**
> *“Appendix lists ‘Impact of Audio.’ Can you quantify modality contributions per reasoning type?”*
>
> **Response:**
> To provide a clearer understanding of how different modalities contribute to scientific video reasoning, we add **audio-only** results for Gemini-2.5-Pro. Below are the modality-specific accuracies comparison, which allow us to disentangle the role of narration (audio) versus visual evidence (video):
>
> | Modality | Overall | Conceptual | Hypothetical | Quantitative |
> |----------|---------|------------|--------------|--------------|
> | **Audio-only** | 34.90 | 38.92 | 39.22 | 22.04 |
> | **Video-only** | 64.30 | 69.73 | 67.79 | 50.61 |
> | **Audio + Video** | **67.00** | **74.32** | **67.53** | **55.10** |
>
> **Analysis:**
> - **Audio-only performance** is significantly above random but far below video-only performance. This indicates that narration provides helpful contextual or procedural cues, especially for **Conceptual** and **Hypothetical** reasoning, but it cannot substitute for actual visual evidence.
> - **Video-only performance** is substantially higher across all categories, confirming that **visual information is the primary modality** required to solve SciVideoBench questions.
> - The modest gain from **Audio + Video** over video alone (e.g., +2.7% overall) shows that **audio augments visual reasoning**, but visual cues remain essential—especially for **Quantitative** reasoning, where measurements and spatial relationships come directly from the video.
> - These results align with our **Transcript-only** and **Visual-blind** results in Weakness 1.4: narration helps but cannot replace visual grounding.
>
> Together, these modality ablations demonstrate that SciVideoBench questions are genuinely **video-dependent**, and that the benchmark evaluates multimodal scientific reasoning in a controlled and interpretable manner.

---

### Official Review · Reviewer_FZjT · 2025-11-01

**Soundness:** 3
**Presentation:** 3
**Contribution:** 3
**Rating:** 4
**Confidence:** 3

**Summary:**

This paper introduces SCIVideoBench, a benchmark for evaluating scientific video reasoning in large multimodal models. It focuses on research-level experimental videos sourced from JoVE and constructs 1,000 multiple-choice questions across four disciplines. The benchmark aims to assess domain-specific, spatiotemporal, and logical reasoning abilities of LMMs, and reports results over 35 models, highlighting substantial performance gaps between proprietary and open-source systems. The work fills a relatively underexplored space between general video QA and specialized scientific reasoning.

**Strengths:**

1. The benchmark addresses an underrepresented domain: research-level scientific experiments, that goes beyond existing video QA datasets focused on everyday scenes or lectures.
2. The dataset construction pipeline (multi-agent annotation + human verification) is well-documented and demonstrates substantial engineering and curation effort.
3. The evaluation over a wide range of open and proprietary models provides a clear empirical baseline for future LMM research in scientific contexts.

**Weaknesses:**

1. The human baseline design is insufficiently described—participants’ domain alignment, access to supporting materials, and the rationale for the closed-book setup remain unclear, raising questions about the interpretability of the 17% human accuracy.
2. Although the text-only baseline provides aggregate evidence for visual dependence, the paper lacks per-item verification (e.g., labeling or statistics on which questions are visually indispensable vs. text-sufficient).

**Questions:**

1. Human baseline: What were the participants’ backgrounds and how closely did they match the disciplines of the videos? Were they allowed to use papers, subtitles, or frame-by-frame playback? What motivated the closed-book design?
2. Answerability: Is there a visual-evidence audit (timestamps, annotated frames, or criteria) verifying that each question truly requires watching the video?

---

> ### Author Response · Authors · 2025-11-23
> **[1/3] Response to Reviewer FZjT**
>
> We thank Reviewer FZjT for the thoughtful evaluation of our work and for recognizing the contribution of SciVideoBench in addressing research-level scientific video reasoning. We appreciate the reviewer’s positive comments on the novelty of the domain, the rigor of the dataset construction process, and the breadth of the evaluation. Below, we provide point-by-point responses to the reviewer’s weaknesses and questions.
>
>
> ## **1. Human baseline design insufficiently described**
>
> **Response:**
> Please see the response to Q1 below.
>
>
> ## **2. Lack of per-item verification of visual dependence**
>
> **Response:**
> Every question in SciVideoBench is explicitly designed to require **visual dependence** on the video. This requirement is enforced at the item level through the structure of the questions themselves:
>
> - All **Conceptual** and **Hypothetical** questions include an explicit **timestamp reference**, which forces both annotators and models to inspect the designated visual segment rather than rely on textual priors alone (as illustrated in Figure 4 of the main paper).
> - All **Quantitative** questions inherently require extracting measurements—such as distances, angles, speeds, counts, or volumetric changes—directly from the video, because **no numerical values are provided in the question prompt**. This ensures that the answer cannot be inferred without careful visual examination.
>
> Thus, by construction, each item is tied to concrete spatiotemporal evidence in the video, guaranteeing per-item visual grounding and preventing text-only shortcuts.

---

> > ### Author Response · Authors · 2025-11-23
> > **[2/3] Response to Reviewer FZjT**
> >
> > ## 3. Response to Reviewer’s Questions
> >
> > ### **Q1. Human Baseline**
> >
> > #### **1. Participants’ backgrounds and alignment with the scientific domains**
> >
> > **Response:**
> > For the human baseline, we recruited **three graduate students per discipline** (Physics, Chemistry, Biology, and Medicine). All participants were advanced graduate students with substantial STEM coursework and laboratory exposure.
> >
> > However, perfect expertise alignment is inherently challenging because SciVideoBench spans **over 25 specialized scientific subjects**, many of which contain deep subfields. For instance, within organic chemistry alone, sub-domains such as spectroscopy, synthesis planning, reaction mechanisms, and chemical biology require very different training. Similarly, in biology, topics range from neurophysiology to molecular genetics to bioengineering.
> >
> > Because no small cohort can perfectly match the expertise required across all these research-grade experimental videos, the baseline reflects the performance of **well-trained but non-specialist graduate students**. This provides a realistic and interpretable reference point for general scientific comprehension, rather than expert-level specialization.
> >
> > #### **2. Were they allowed to use papers, subtitles, or frame-by-frame playback?**
> >
> > **Response:**
> > Participants were provided **only the videos, questions, and answer options**—no subtitles, JoVE papers, textbooks, internet search, or external materials were permitted. The evaluation was conducted in a **closed-book setting** to ensure that answers were grounded strictly in the visual and procedural evidence in the video rather than external domain knowledge.
> >
> > They were free to **rewatch and replay the videos as many times as they wished**, including pausing and reviewing fine-grained motions, which mirrors realistic scientific video comprehension.
> >
> > #### **3. What motivated the closed-book design?**
> >
> > **Response:**
> > The closed-book setup was intentionally chosen to ensure that the human evaluation reflects **pure video-based scientific reasoning**, rather than reliance on external textual knowledge. SciVideoBench is designed to assess multimodal capabilities such as observation, temporal tracking, procedural understanding, and quantitative interpretation. Allowing access to supplementary materials (e.g., JoVE papers, subtitles, or external references) would shift the task toward text-driven scientific QA, undermining the benchmark’s core objective of evaluating reasoning grounded directly in the **visual evidence**.
> >
> > The closed-book protocol also ensures a **fair and consistent comparison** between humans and models, since models are evaluated solely based on video input without access to additional scientific documents unless explicitly specified in an ablation.
> >
> > Furthermore, the observed human performance supports our central hypothesis:
> > SciVideoBench requires **research-level, domain-specific expertise** that goes beyond the general background typically possessed by graduate students in each discipline. The low human accuracy does not indicate poor calibration; rather, it highlights the benchmark’s difficulty and its emphasis on complex experimental reasoning, which cannot be solved through common textbook knowledge characteristic of college-level benchmarks.
> >
> > Overall, the closed-book design preserves the integrity of the benchmark and accurately reflects the scientific reasoning demands that SciVideoBench aims to evaluate.

---

> > > ### Author Response · Authors · 2025-11-23
> > > **[3/3] Response to Reviewer FZjT**
> > >
> > > ### **Q2. Answerability: Is there a visual-evidence audit verifying that each question truly requires watching the video?**
> > >
> > > **Response:**
> > > Yes. SciVideoBench incorporates multiple layers of verification to ensure that every question genuinely requires **visual and temporal evidence** from the video.
> > >
> > > First, by construction, the question design protocol ensures that **no item can be answered without visual input**. Conceptual and hypothetical questions include explicit **timestamp references**, while quantitative questions inherently require extracting measurements from the video, as no numerical information is provided in the prompt.
> > >
> > > Beyond question design, SciVideoBench enforces a rigorous **visual-evidence and timestamp audit** throughout the annotation pipeline:
> > >
> > > 1. **Timestamp grounding validation**
> > >    - Human annotators verify that every referenced action, motion, scientific phenomenon, or procedural step appears **exactly** within the indicated timestamp range.
> > >    - Any misalignment or hallucinated reference leads to immediate rejection or mandatory correction.
> > >
> > > 2. **Answerability checking**
> > >    - Annotators ensure that each question is answerable **exclusively from the video**, without relying on external scientific facts, textbook knowledge, or high-level domain assumptions.
> > >    - If a question requires information not observable in the video, it is removed from the dataset.
> > >
> > > Together, these steps constitute a strict **visual-evidence audit**, guaranteeing that all SciVideoBench items demand genuine spatiotemporal and experimental reasoning rather than textual shortcuts or prior-knowledge exploitation.

---

### Official Review · Reviewer_aWZV · 2025-11-01

**Soundness:** 3
**Presentation:** 3
**Contribution:** 3
**Rating:** 6
**Confidence:** 4

**Summary:**

This paper introduces SciVideoBench, a new benchmark designed to evaluate advanced video reasoning capabilities of Large Multimodal Models (LMMs) in scientific contexts. The benchmark consists of 1,000 multiple-choice (more than 4 choices) questions derived from 241 research-level experimental videos from the Journal of Visualized Experiments (JoVE), spanning four major disciplines (Physics, Chemistry, Biology, Medicine) and over 25 specialized subjects. Questions are categorized into three types: conceptual, hypothetical, and quantitative reasoning. The authors employ a semi-automatic annotation pipeline combining multi-agent LLM systems with human expert verification. Evaluation of 35 models (5 proprietary, 30 open-source) reveals significant performance gaps, with the best model (Gemini 2.5 Pro) achieving only 64.3% accuracy, and best open-source model (InternVL-3-78B) only achieves 38.0, indicating substantial room for advancement in scientific video understanding.

**Strengths:**

1. The benchmark addresses a critical gap in evaluating LMMs on research-level scientific reasoning, moving beyond elementary/college-level content that current models are beginning to saturate.
2. The semi-automatic annotation pipeline with human-in-the-loop verification, multiple specialized agents (QA Generator, Evaluator, Visual Comparer, Refiner), and expert validation ensures high-quality questions that require genuine multimodal reasoning.
3. The paper evaluates 35 models across multiple dimensions (proprietary vs. open-source, different sizes, with/without chain-of-thought prompting) and provides detailed analysis of failure modes. And the benchmarks shows significant gap between close-source and open-source models' performance, highlighting a important direction for the open-source community to push.

**Weaknesses:**

1. The paper lacks one doable ablation that whether the performance of models scale as the number of frames sampled scaled up. This is also essential for a video understanding benchmark.
2. The low human performance makes it difficult to assess whether questions are genuinely research-level or simply poorly calibrated

**Questions:**

1. Given that Gemini 2.5 Pro was used extensively in annotation, were any steps taken to validate that the benchmark doesn't inadvertently favor Gemini models? What would performance look like if a different model had been used for annotation? Is the high performance of Gemini 2.5 Pro on the benchmark because of this issue?
2. What metrics were used to validate annotation quality beyond human review? Was inter-annotator agreement measured? How many questions were rejected during the validation process?

---

> ### Author Response · Authors · 2025-11-22
> **[1/4] Response to Reviewer aWZV**
>
> We thank Reviewer aWZV for the constructive feedback and thoughtful evaluation of our submission. We appreciate the recognition of SciVideoBench’s contributions, including its focus on research-level scientific reasoning, the rigor of the multi-agent annotation pipeline, and the comprehensive evaluation across a wide range of open-source and proprietary models.
>
> Below, we provide point-by-point responses to the reviewer’s comments.
>
>
> ## **1. Ablation on frame sampling scale**
>
> For fairness and consistency, we follow each model’s **official default frame-sampling strategy** during evaluation. Because SciVideoBench contains visually dense experimental procedures, increasing the number of frames naturally provides **more visual evidence**, which should help models better track temporal dependencies, observe fine-grained experimental details, and reason more accurately about the underlying scientific procedures.
>
> To validate this hypothesis, we conducted additional ablation experiments on **Qwen2.5-VL (7B and 3B)** using multiple frame-sampling settings. The results below clearly show that, for both model sizes, in general, performance **drops consistently as fewer frames are provided**, confirming that SciVideoBench requires substantial visual information and is sensitive to temporal coverage.
>
> | Model & Frames | Overall | Conceptual | Hypothetical | Quantitative|
> | -------------- | -------- | ---------- | ------------ | ------------ |
> | Qwen2.5-VL-7B |  |  | |  |
> | **768f**  | **26.3** | **35.95**      | **28.05**        | 8.98        |
> | **128f** | 25.8 | 34.86 | 25.97 | 11.84 |
> | **64f** | 25.5 | 33.78 | 26.23 | 11.84 |
> | **32f**   | 25.4     | 35.14      | 24.42        | **12.24**        |
> | **16f** | 22.2 | 29.19 | 22.34 | 11.43 |
> | Qwen2.5-VL-3B |  |  | |  |
> | **768f**  | **22.5** | **32.43**      | **20.78**        | **10.20**        |
> | **128f** | 22.1 | 31.89 | 20.26 | **10.20** |
> | **64f** | 20.7 | 31.08 | 17.66 | 8.98 |
> | **32f** | 20.9 | 29.46 | 19.48 | **10.20** |
> | **16f** | 19.9 | 27.84 | 18.96 | 9.39 |
>
> These ablations confirm that SciVideoBench rewards models that can effectively leverage **rich temporal visual signals**, and that higher frame sampling provides clear benefits for scientific video reasoning.

---

> ### Author Response · Authors · 2025-11-22
> **[2/4] Response to Reviewer aWZV**
>
> ## **2. Low human performance and question difficulty calibration**
>
> We appreciate the reviewer’s concern regarding the interpretation of human performance. We address this in two parts: (1) the reliability of our annotation quality, and (2) why lower human performance reflects the genuine difficulty of research-level scientific video reasoning rather than poor question calibration.
>
> #### **a. Annotation quality is rigorously ensured through multi-stage expert verification**
>
> Our annotation pipeline integrates multiple levels of human and expert validation to guarantee grounding quality and consistency. Five annotators with **PhD-level training** and prior experience in STEM research or scientific video analysis participated in the verification process. Although we did not report a numerical inter-annotator agreement score in the original submission, our workflow incorporates *systematic agreement checks* through repeated cross-validation and structured review.
>
> Each question–answer pair is evaluated along two critical dimensions:
>
> - **Answerability:** ensuring that the question can be answered *solely* from the visual and scientific evidence in the video, without requiring external knowledge.
> - **Temporal grounding:** verifying that all referenced actions, operations, or quantities appear exactly within the specified timestamp range.
>
> Every annotation batch undergoes a **multi-pass quality check**, where a second annotator re-validates the items and any disagreement triggers a structured resolution step involving expert review. This procedure serves as a practical surrogate for formal inter-annotator agreement and ensures consistently high-quality annotations throughout the benchmark.
>
> #### **b. Lower human performance reflects the genuine research-level difficulty of SciVideoBench**
>
> The human baseline was designed to reflect the capabilities of **well-trained but non-specialist graduate students**, a realistic and meaningful comparator for assessing general scientific comprehension. We recruited **three graduate students per discipline (Physics, Chemistry, Biology, Medicine)**, all of whom answered the questions in a strict **closed-book** setting. Only the **videos, questions, and answer options** were provided, and participants were allowed to freely replay the videos within a single session.
>
> The breadth and depth of SciVideoBench inherently pose significant challenges: the benchmark spans **over 25 specialized subjects**, many of which contain deeply technical subfields. **Even within a single domain such as organic chemistry, sub-areas like reaction mechanisms, spectroscopy, synthesis, or chemical biology require distinct expertise. It is therefore *not feasible* to assemble a human evaluation cohort whose training fully matches every specialized research topic represented in the dataset.**
>
> The resulting lower human accuracy is thus an expected and meaningful reflection of the **research-level difficulty** of the benchmark rather than an indication of poor question calibration. In fact, it underscores the challenge of requiring LMMs to perform authentic scientific video reasoning that demands observation, measurement, multi-step reasoning, and quantitative calculation grounded in real laboratory procedures.
>
> In summary, the annotation quality is rigorously controlled through multi-stage expert validation, and the lower human performance is a natural outcome of the benchmark’s research-level complexity rather than calibration issues.

---

> > ### Author Response · Authors · 2025-11-22
> > **[3/4] Response to Reviewer aWZV**
> >
> > ## 3. Response to Reviewer’s Questions
> >
> >
> > ### **Question 1: Concern about Gemini-2.5-Pro influence on annotation**
> >
> > #### **1.1 Were any steps taken to validate that the benchmark does not inadvertently favor Gemini models?**
> >
> > **Response:**
> > Yes. Although Gemini-2.5-Pro was used during early annotation to assist with timestamp extraction and dense captioning, its role was strictly **supportive rather than decisive**. Every LLM-generated output—timestamps, captions, and draft QAs—underwent **rigorous multi-stage human validation**, ensuring that no model-specific bias propagates into the final benchmark.
> >
> > Specifically:
> >
> > - All questions were required to pass a **visual-blind test using GPT-4o**, ensuring that the answer could not be derived from linguistic artifacts introduced by Gemini.
> > - Every question also underwent **human re-verification** for answerability and timestamp grounding.
> > - Only items that passed **both** the automated visual-blind filtering and the human verification stages were kept in the final dataset.
> >
> > This pipeline ensures that Gemini-2.5-Pro does **not** shape the correct answers, question style, or reasoning structure of SciVideoBench.
> >
> > ---
> >
> > #### **1.2 What would performance look like if a different model had been used for annotation?**
> >
> > **Response:**
> > Gemini-2.5-Pro was chosen because, at the time we began constructing SciVideoBench, it was the **only model capable of reliably producing accurate timestamp-aligned descriptions** for research-grade experimental videos. However, it was not the only model used in the pipeline:
> >
> > - **GPT-4o** was used for the **visual-blind test** during the annotation process, option design verification.
> > - Human annotators performed multi-stage validation to eliminate errors, inconsistencies, or stylistic patterns from Gemini.
> >
> > Thus, even if another model had been used, the final content would remain largely unchanged, because it is **human experts determined the final question, timestamp, and answer validity**. Gemini-2.5-Pro provides valuable assistance in the pipeline, but its role is supportive rather than decisive.
> >
> > ---
> >
> > #### **1.3 Is the high performance of Gemini-2.5-Pro on the benchmark caused by this issue?**
> >
> > **Response:**
> > No. We explicitly evaluated **GPT-5** and **Gemini-3-Pro**, both of which outperform Gemini-2.5-Pro on quantitative reasoning. Gemini-3-Pro achieves the highest overall accuracy and consistently surpasses Gemini-2.5-Pro across all categories.
> >
> > This clearly demonstrates that SciVideoBench does **not** favor Gemini-2.5-Pro.
> >
> > | Models            | Overall | Conceptual | Hypothetical | Quantitative |
> > |-------------------|---------|------------|--------------|--------------|
> > | Gemini-2.5-Pro    | 64.30   | 69.73      | 67.79        | 50.61        |
> > | GPT-5             | 55.70   | 61.35      | 50.65        | **55.10**    |
> > | **Gemini-3-Pro**  | **70.40** | **76.22**  | **72.21**    | **58.78**    |

---

> > > ### Author Response · Authors · 2025-11-22
> > > **[4/4] Response to Reviewer aWZV**
> > >
> > > ### **Question 2: Annotation quality metrics and inter-annotator agreement**
> > >
> > > #### **2.1 What metrics were used to validate annotation quality beyond human review?**
> > >
> > > **Response:**
> > > In addition to human verification, we used **option-only guessing** and **visual-blind testing** as diagnostic checks to ensure annotation reliability and eliminate linguistic artifacts.
> > >
> > > Below are the **option-only guess results** using GPT-4o:
> > >
> > > | Mode              | Overall | Conceptual | Hypothetical | Quantitative |
> > > |-------------------|---------|------------|--------------|--------------|
> > > | Option-only Guess | 10.5    | 11.35      | 10.65        | 8.98         |
> > >
> > > These results are close to **pure random guessing under 10-option MCQA**, confirming that:
> > >
> > > - the option design avoids knowledge leakage,
> > > - there is no shortcut through language artifacts,
> > > - and the correct answers genuinely depend on the video content.
> > >
> > > ---
> > >
> > > #### **2.2 Was inter-annotator agreement measured?**
> > >
> > > **Response:**
> > > Although we did not compute a numerical inter-annotator agreement score, our pipeline incorporates a **systematic, multi-pass agreement mechanism**:
> > >
> > > - Every annotation batch is **re-checked by a second annotator**.
> > > - Any disagreement triggers a **structured resolution process** involving expert review.
> > > - Verification covers both **timestamp correctness** and **answerability**.
> > >
> > > This procedure acts as a **practical surrogate for formal inter-annotator agreement**, ensuring consistency and grounding quality across the entire benchmark.
> > >
> > > ---
> > >
> > > #### **2.3 How many questions were rejected?**
> > >
> > > **Response:**
> > > The initial LLM-assisted generation produced **more than 4,000 draft questions**.
> > > Over **3,000 questions** were rejected through:
> > >
> > > - the GPT-4o visual-blind test,
> > > - multi-stage human validation,
> > > - timestamp cross-verification,
> > > - answerability checks.
> > >
> > > Only questions that passed **all** validation steps were included in the final **1,000-question** benchmark.

---

### Official Review · Reviewer_wpTT · 2025-11-01

**Soundness:** 3
**Presentation:** 3
**Contribution:** 3
**Rating:** 6
**Confidence:** 3

**Summary:**

This paper introduces **SciVideoBench**, a benchmark designed to evaluate advanced video reasoning capabilities of Large Multimodal Models (LMMs) in scientific contexts. The benchmark consists of 1,000 multiple-choice questions derived from 241 research-grade experimental videos sourced from the Journal of Visualized Experiments (**JoVE**), spanning over 25 scientific subjects across four major disciplines. Each question is categorized as conceptual, hypothetical, or quantitative, requiring sophisticated domain knowledge, precise spatiotemporal perception, and complex logical reasoning. The authors evaluate 35 models including proprietary systems (Gemini 2.5 Pro, GPT-4o) and open-source alternatives, revealing significant performance gaps with the best models achieving only modest accuracy compared to human experts. The work aims to address limitations in existing video benchmarks that have become saturated and fail to adequately challenge advanced multimodal reasoning capabilities.

**Strengths:**

1. **Novel and challenging benchmark addressing a critical gap**
- Targets research-level scientific reasoning rather than college-level knowledge, filling a significant void in current evaluation frameworks (Sec. 1, Table 1)
- Demonstrates clear performance gaps even for state-of-the-art models like Gemini 2.5 Pro (≈27% accuracy vs. 64% human performance), indicating substantial room for improvement (Table 2)
2. **Comprehensive experimental design and evaluation methodology**
- Evaluates 35 diverse models across proprietary and open-source categories with consistent experimental protocols (Sec. 4)
- Implements multi-stage human-AI collaborative annotation pipeline with quality control mechanisms (Sec. 3.2, Figure 2)
- Provides detailed analysis across question types (conceptual, hypothetical, quantitative) revealing differential model capabilities (Sec. 4.3, Figure 5)
3. **High-quality dataset construction with rigorous validation**
- Covers broad scientific scope with 25+ subjects across Biology, Chemistry, Physics, and Engineering ensuring domain diversity (Figure 3)
- Provides clear categorization framework distinguishing conceptual understanding, hypothetical reasoning, and quantitative analysis (Sec. 3.2)
4. **Practical impact and reproducibility considerations**
- Addresses real-world scientific video understanding needs relevant to AI for Science applications (Sec. 2.2)
- Provides detailed error analysis revealing specific failure modes like ignoring visual evidence and missing domain knowledge (Sec. 4.4)

**Weaknesses:**

1. **Dataset construction and annotation concerns**
- Human annotation process details are underspecified, particularly regarding inter-annotator agreement and quality control metrics (Sec. 3.2)
- No reported analysis of potential annotation bias or consistency across different scientific domains (Sec. 3.2)
- Limited discussion of how question difficulty was calibrated or validated beyond expert review (Sec. 3.2)
2. **Limited theoretical analysis of reasoning requirements**
- Lacks formal characterization of what constitutes “research-level” reasoning versus college-level knowledge, relying primarily on intuitive distinctions (Sec. 1)
- No systematic analysis of cognitive complexity or reasoning depth required for different question types beyond basic categorization (Sec. 3.2)
3. **Methodological and statistical rigor issues**
- No reported confidence intervals or statistical significance testing for performance comparisons across models (Table 2)
- Missing analysis of question difficulty distribution and potential ceiling/floor effects in evaluation (Sec. 4)
- Limited discussion of potential dataset contamination or overlap with model training data (Sec. 4)

**Questions:**

How do you formally define and validate “research-level” reasoning? Can you provide cognitive complexity metrics or expert assessments that distinguish your questions from existing college-level benchmarks? What specific reasoning skills (causal inference, counterfactual reasoning, multi-step deduction) are required?

---

> ### Author Response · Authors · 2025-11-22
> **[1/4] Response to Reviewer wpTT**
>
> We sincerely thank Reviewer **wpTT** for the thoughtful and detailed review, and we greatly appreciate the recognition of the novelty, significance, and rigorous design of **SciVideoBench**. Below, we respond to each weakness and question point-by-point.
>
> ## 1. Dataset Construction and Annotation Concerns
>
> ### 1.1 Inter-annotator agreement and quality control metrics
> Our annotation pipeline includes multiple levels of human and expert validation to ensure consistency and grounding quality. Specifically, five human annotators with PhD-level training participated in the verification process, all with experience in scientific video analysis or related STEM domains. While we did not explicitly report a numerical inter-annotator agreement metric in the original submission, the annotation workflow incorporates systematic agreement checks through repeated cross-validation and structured verification stages.
>
> During annotation refinement, annotators evaluate each question–answer pair along two key dimensions:
>
> - Answerability: ensuring that each question can be answered exclusively based on the visual and scientific evidence present in the video. Annotators confirm that no outside domain knowledge is required beyond what is observable or directly implied by the experiment.
>
> - Temporal grounding: verifying that all referenced actions, procedures, motions, or quantities appear precisely within the stated timestamp range. Annotators check alignment between the question, the correct answer, and the specific visual segment to avoid hallucinated or incorrectly localized evidence.
>
> Consistency is further enforced through iterative quality checks: each batch of annotations is re-evaluated by a second annotator, and disagreements trigger a structured resolution step involving expert review. This multi-pass process serves as a practical surrogate for formal inter-annotator agreement metrics and ensures high-quality grounding across the dataset.
>
> ### 1.2 Potential annotation bias across scientific domains
> We appreciate the reviewer’s concern regarding potential annotation bias across different scientific domains or question types. In practice, several aspects of our annotation pipeline help minimize such bias.
>
> First, Gemini-2.5-Pro is used to generate **dense captions** that consistently capture the core scientific aspects of each video across three fundamental dimensions:
> 1. **Scientific principle** behind the experiment (Conceptual reasoning)
> 2. **Experimental operation or procedural step** (Hypothetical reasoning)
> 3. **Scientific calculation or quantitative measurement** (Quantitative reasoning)
>
> These three dimensions map directly to our question types, ensuring balanced coverage and reducing the likelihood that certain subjects receive lower-quality annotations.
>
> Second, our **human verification** process applies domain-invariant checks such as:
> - **Timestamp correctness**
> - **Answerability strictly from the visual evidence**
> - **Clarity and grounding in the scientific procedure**
>
> Because these criteria focus on visual grounding and temporal alignment rather than deep subject-specific expertise, the verification process helps maintain consistent annotation quality across all scientific domains.
>
> Third, we employ **multiple PhD-level annotators** with broad STEM backgrounds. While no individual expert can specialize in all 25+ subjects represented in SciVideoBench, the verification steps (timestamp alignment, scientific plausibility, and visual grounding) are largely independent of subdiscipline specialization. This helps reduce potential domain-specific annotation bias.
>
> Overall, the combination of multimodal caption generation, domain-agnostic human validation, and cross-disciplinary expertise keeps annotation bias across subjects and question types minimal. We will clarify this explicitly in the revised version.
>
> ### 1.3 Question Difficulty Calibration
> In our annotation pipeline, we did not explicitly assign a difficulty label to each question, as the research-level protocol inherently ensures that all questions in SciVideoBench require either advanced domain expertise, multi-step scientific reasoning, or complex quantitative analysis. Nonetheless, the difficulty characteristics naturally emerge through the three question types, and this pattern is reflected clearly in the evaluation results. As shown in the main table, **quantitative reasoning consistently yields the lowest accuracy across all models**, indicating that it represents the most challenging category. For conceptual and hypothetical reasoning, proprietary models tend to exhibit similar performance levels, whereas most open-source models perform noticeably better on **conceptual** questions—likely due to their closer alignment with textbook-style scientific knowledge. This empirical trend aligns with our intended design and confirms that question difficulty is meaningfully differentiated across reasoning types.

---

> ### Author Response · Authors · 2025-11-22
> **[2/4] Response to Reviewer wpTT**
>
> ## 2. Limited Theoretical Analysis of Reasoning Requirements
>
> ### 2.1 Formal characterization of “research-level” reasoning
>
> We thank the reviewer for raising this point. In the paper, we contrasted college-level and research-level reasoning at a high level, but we agree that a more formal definition clarifies the distinction. Below, we provide a precise characterization tailored to SciVideoBench.
>
> We define **research-level scientific reasoning** as the cognitive process required to interpret, integrate, and infer insights from *real experimental evidence*, *procedural context*, and *scientific mechanisms* presented in authentic laboratory environments. A question in SciVideoBench requires research-level reasoning if solving it necessitates one or more of the following:
>
> 1. **Experimental-procedure reasoning:** inferring causal relations, operational constraints, or underlying scientific principles from observed laboratory actions rather than textbook descriptions.
> 2. **Counterfactual or hypothetical reasoning:** predicting how experimental outcomes change under modified conditions based on domain principles.
> 3. **Quantitative and measurement-based reasoning:** interpreting numeric readings (e.g., volumes, times, distances) from instruments in the video and applying appropriate scientific equations.
> 4. **Multi-step visual–scientific deduction:** connecting sequential events in the video into a coherent chain of reasoning involving scientific mechanisms.
> 5. **Integration of multimodal scientific evidence:** jointly using visual observations, procedural cues, and narration to infer scientific intent or mechanisms.
>
> This differs fundamentally from **college-level questions**, which typically rely on idealized textbook content, formula recall, or simplified numerical exercises. In contrast, SciVideoBench questions require reasoning over non-ideal experimental conditions, real procedural workflows, and visual cues that cannot be solved without multimodal grounding.
>
> We will include this formal definition in the revised version to address the reviewer’s concern directly.
>
>
> ### 2.2 Systematic analysis of reasoning depth
>
> To systematically characterize the cognitive complexity required by SciVideoBench, our question generation pipeline incorporates **explicit reasoning-chain generation** using Gemini-2.5-Pro. For every question, the model produces a detailed, step-by-step reasoning process. This reasoning chain is then **manually verified by human annotators** to ensure (1) factual correctness, (2) temporal alignment with visual evidence in the video, and (3) consistency with scientific principles. This procedure ensures that the underlying reasoning structure is both scientifically valid and visually grounded.
>
> To quantitatively assess reasoning depth, we compute two cognitive-complexity metrics across all questions:
> - the **average number of reasoning steps**
> - the **average number of reasoning words**
>
> These metrics provide a principled, model-independent measurement of the reasoning difficulty inherent in each question type. The aggregated statistics are:
>
> | Question Type | Avg. Reasoning Steps | Avg. Reasoning Words |
> |---------------|-----------------------|------------------------|
> | **Conceptual**    | 5.58                  | 101.56                 |
> | **Hypothetical**  | 4.78                  | 91.00                  |
> | **Quantitative**  | 10.59                 | 342.63                 |
>
> These results demonstrate that all question types require multi-step reasoning. Notably, **quantitative questions** demand substantially longer and more complex inference chains, involving detailed interpretation of measurements, numerical calculations, and instrument readings. This aligns with our formal definition of research-level scientific reasoning and highlights the deeper cognitive demands inherent in quantitative items.
>
> Together, these reasoning-chain metrics provide a rigorous and systematic characterization of reasoning depth, directly addressing the reviewer’s concern regarding complexity analysis. **Moreover, they offer a consistent explanation for the observed difficulty patterns discussed in Response 1.3, where quantitative questions exhibit the greatest challenge across models.**

---

> ### Author Response · Authors · 2025-11-22
> **[3/4] Response to Reviewer wpTT**
>
> ## 3. Methodological and Statistical Rigor
>
> ### 3.1 Statistical significance and confidence intervals
> Our evaluation uses **temperature = 0** for all models to ensure deterministic outputs, which eliminates stochastic variation and significantly reduces the need for repeated trials or variance estimation. This setting is standard in multimodal QA benchmarks, as it ensures that any performance differences reflect true model capability rather than sampling randomness.
>
> Additionally, SciVideoBench includes a **large number of evaluated models**, spanning both proprietary and open-source systems. Running full repeated inferences for every model would introduce substantial computational and financial overhead, making statistical resampling impractical at this scale. Instead, we focus on consistent experimental settings and controlled inputs to ensure fairness across comparisons.
>
> ### 3.2 Difficulty Distribution and Ceiling/Floor Effects
> As discussed in Responses 1.3 and 2.2, the difficulty of SciVideoBench questions naturally emerges from the research-level annotation protocol and the inherent reasoning complexity within each question type. Although we do not explicitly assign difficulty labels to individual questions, the empirical performance patterns across conceptual, hypothetical, and quantitative reasoning clearly demonstrate meaningful differentiation in difficulty.
>
> To further analyze potential **ceiling and floor effects**, we additionally report **option-only** and **video+paper** performance using GPT-4o. We view the option-only condition as a theoretical lower bound beyond visual-blind testing, because it isolates pure guessing under the 10-option MCQA setting without the question. Conversely, the video+paper condition represents an approximate upper bound: each JoVE video is paired with a published paper, which provides additional methodological details and scientific context that a model could leverage. The gap between these baselines and full video performance illustrates that SciVideoBench neither suffers from severe floor effects nor approaches a performance ceiling for strong models.
>
> | Mode            | Overall | Conceptual | Hypothetical | Quantitative |
> |-----------------|---------|------------|--------------|--------------|
> | Option-only Guess     | 10.5   | 11.35      | 10.65        | 8.98        |
> | Video QA (from the main table)      | 24.90   | 30.27      | 28.05        | **11.84**        |
> | Video+Paper QA | **25.80** | **30.81**  | **30.13**    | 11.43        |
>
>
> ### 3.3 Dataset contamination/training overlap
> We designed SciVideoBench to minimize the likelihood of dataset contamination or overlap with existing model training data. First, many of the JoVE articles used in our benchmark come from the **most recent half-year**, published **after April 2025**. These materials cannot be included in the training data of current state-of-the-art models. For example, Gemini 2.5 Pro has a knowledge cutoff of **January 2025**, and GPT-4o has a cutoff of **August 2024**, making it impossible for these models to have seen the newer JoVE content used in SciVideoBench.
>
> Looking forward, we plan to maintain SciVideoBench as a **continuously updated benchmark**. By refreshing and expanding the dataset each year with newly published JoVE videos, we can ensure that future models cannot exploit training-data overlap. This will help preserve the benchmark’s integrity, prevent cheating through memorization, and provide a reliable evaluation of true scientific video reasoning capabilities.

---

> ### Author Response · Authors · 2025-11-22
> **[4/4] Response to Reviewer wpTT**
>
> ## 4. Response to Reviewer’s Question
>
> **Question 1: How do you formally define and validate ‘research-level’ reasoning?**
>
> **Response:**
> Please see the response 2.1.
>
> **Question 2: Can you provide cognitive complexity metrics or expert assessments that distinguish your questions from existing college-level benchmarks?**
>
> **Response:**
> We refer the reviewer to the **cognitive complexity analysis in Response 2.2**, where we quantify the reasoning depth.
>
> For comparison, MMVU [1] also provides rationales, but these rationales generally consist of a single observation paired with a direct factual recall or a one-step knowledge application. They do not provide structured multi-step reasoning chains. Qualitatively, this reflects a notable gap in cognitive depth: SciVideoBench requires multi-stage inference situated within real laboratory procedures, whereas MMVU rationales tend to be closer to a single inference step.
>
> As mentioned earlier, we do not provide an explicit *expert-assigned difficulty label* for each question to directly compare with college-level benchmarks such as MMVU [1]. However, our annotation protocol imposes systematic and stringent requirements: each question must challenge the model with **research-level scientific reasoning** in real-world experimental scenarios. Models must be able to **observe**, **measure**, **reason**, and **calculate** based on dynamic visual evidence from actual laboratory videos. This is fundamentally different from benchmarks like MMVU, where many questions rely primarily on domain facts from Wikipedia or standard educational knowledge.
>
> Together, the multi-step reasoning metrics, real-world experimental grounding, and scientific procedural dependencies provide a strong distinction between SciVideoBench and existing college-level benchmarks.
>
> [1] Yilun Zhao, Lujing Xie, Haowei Zhang, Guo Gan, Yitao Long, Zhiyuan Hu, Tongyan Hu, Weiyuan Chen, Chuhan Li, Junyang Song, Zhijian Xu, Chengye Wang, Weifeng Pan, Ziyao Shangguan, Xiangru Tang, Zhenwen Liang, Yixin Liu, Chen Zhao, Arman Cohan.  MMVU: Measuring Expert-Level Multi-Discipline Video Understanding, 2025.
>
>
> **Question 3: What specific reasoning skills (causal inference, counterfactual reasoning, multi-step deduction) are required?**
> **Response:**
> As mentioned in Question 2, SciVideoBench is designed explicitly to assess *research-level scientific reasoning*, which goes far beyond factual recall or single-step interpretation. The benchmark requires multiple forms of higher-order reasoning that naturally arise in real experimental workflows. Specifically, the questions engage the following reasoning skills:
>
> 1. **Causal Inference:**
>    Many experimental procedures involve cause–effect relationships, such as how modifying a reagent concentration, altering electrode placement, or adjusting temperature influences the observed outcome. Models must identify causal chains from dynamic visual evidence, not from static textual descriptions.
>
> 2. **Counterfactual and Hypothetical Reasoning:**
>    Hypothetical questions frequently ask what would happen *if one step were omitted*, or *if an instrument behaved differently*. These require reasoning over unseen but scientifically plausible scenarios grounded in the experimental setup.
>
> 3. **Multi-Step Deduction:**
>    Quantitative questions require multi-stage computation—extracting measurements from video frames, interpreting the meaning of those measurements, recalling relevant scientific formulas, and performing multi-step calculations. This is reflected in the reasoning-chain statistics provided in Response 2.2, where quantitative questions show the longest and most complex inference chains.
>
> 4. **Procedural Reasoning and Temporal Understanding:**
>    Many questions require tracking the temporal progression of an experiment—identifying which step occurs when, understanding dependencies between steps, and deducing why a particular operation is required. This is a key distinguishing factor from college-level benchmarks, which rarely require fine-grained temporal reasoning grounded in real-world actions.
>
> 5. **Cross-Modal Scientific Integration:**
>    Models must integrate visual signals (instrument readings, motions, chemical color changes, equipment settings) with scientific knowledge (experimental principles, safety constraints, physics or chemistry laws). This integration is central to research-level reasoning, where evidence is multimodal by nature.
>
> In combination, these reasoning skills demonstrate that SciVideoBench questions require substantially deeper cognitive processing than those in existing college-level benchmarks. They reflect the authentic reasoning demands of real scientific experiments rather than textbook-style knowledge recall.

---

### Author Response · Authors · 2025-11-22
**[1/3] General Response**

Dear Reviewers, Area Chairs, and Program Chairs,

We sincerely thank all reviewers for the time and thoughtful effort dedicated to evaluating our submission. We greatly appreciate the constructive feedback and are encouraged by the consistent recognition of **SciVideoBench’s contributions** across multiple reviewers. Your comments have highlighted both the significance of the problem we address and the impact of our benchmark on advancing scientific video reasoning in LMMs. Below, we summarize the key strengths noted by the reviewers.


**Motivation, Importance, and Significance (Reviewers wpTT, aWZV, FZjT, 1k1D)** All reviewers emphasized that **SciVideoBench addresses a critical and previously underexplored gap** in evaluating LMMs on research-level scientific video reasoning—far beyond everyday video QA or college-level material. Reviewer **wpTT** highlighted that it covers advanced scientific reasoning across **25+ subjects** in Biology, Chemistry, Physics, and Engineering. Reviewer **aWZV** pointed out that existing video benchmarks are becoming saturated and **fail to challenge modern LMMs**. Reviewer **FZjT** emphasized that SciVideoBench fills *“an underrepresented domain.”* Reviewer **1k1D** noted the work’s high potential impact for multimodal AI in scientific research contexts.


**Novelty of the Scientific Video Reasoning Benchmark  (Reviewers wpTT, aWZV, FZjT, 1k1D)** Reviewers consistently recognized SciVideoBench as the **first benchmark targeting research-grade scientific experiments**, requiring the integration of: multimodal visual evidence, scientific procedural understanding, and domain knowledge reasoning. Reviewer **1k1D** emphasized its **tri-modal grounding** (video + narration + procedures), while reviewer **aWZV** noted that it challenges models with **conceptual, hypothetical, and quantitative reasoning**, going beyond typical recognition tasks.


**High-Quality Dataset Construction and Annotation Pipeline  (Reviewers wpTT, aWZV, FZjT, 1k1D)** All reviewers acknowledged the rigor of our **multi-agent, human-in-the-loop annotation pipeline**. Reviewer **wpTT** praised the multi-stage collaborative design and quality control. Reviewer **aWZV** highlighted the use of specialized agents — *QA Generator, Evaluator, Visual Comparer, Refiner* — paired with expert verification to enforce strict grounding. Reviewer **1k1D** noted the importance of **timestamp alignment and visual validation**. Reviewer **FZjT** recognized the significant **engineering effort** required to curate such a dataset.


**Comprehensive Evaluation and Experimental Design  (Reviewers wpTT, aWZV, FZjT, 1k1D)** Reviewers responded positively to our **extensive evaluation across 35 proprietary and open-source models**. Reviewer **wpTT** described the evaluation as **comprehensive**. Reviewer **aWZV** appreciated the diversity of baselines (CoT prompting, model scaling, proprietary vs. open-source). Reviewer **1k1D** pointed out the inclusion of **vision-blind text-only baselines**, which isolate multimodal contributions.


**Clear Empirical Insights and Failure Analysis  (Reviewers wpTT, aWZV, FZjT, 1k1D)** Reviewers agreed that our results and analyses provide **meaningful insights into the limitations of current LMMs**. Reviewer **wpTT** highlighted the **large performance gaps** between models and humans, noting that even top proprietary models achieve only ~27% accuracy. Reviewer **aWZV** emphasized the analysis of failure modes and the open-source vs. proprietary performance gap. Reviewer **1k1D** mentioned strong **Chain-of-Thought effects** and the role of visual grounding. Reviewer **FZjT** appreciated the **clear empirical baselines** that will guide subsequent research.

In response to your valuable comments, we conducted several additional experiments, including the evaluation results of most recent GPT-5 and Gemini-3-Pro, modality-specific ablations, reasoning-depth analyses, and comprehensive frame-sampling studies. These enhancements have been incorporated into both the main manuscript and the appendix, further strengthening the clarity, rigor, and completeness of SciVideoBench.

---

> ### Author Response · Authors · 2025-11-22
> **[2/3] General Response**
>
> # Clarifications, Additional Experiments, and Evidence Supporting SciVideoBench
>
> ## 1. SciVideoBench Does Not Favor Gemini-2.5-Pro
>
> We acknowledge the reviewer’s concern regarding the use of Gemini-2.5-Pro in the annotation workflow. Importantly, while Gemini-2.5-Pro provides valuable assistance in the pipeline, its role is **supportive rather than decisive**. It helps generate dense, scientifically grounded captions and contributes to identifying the core research problem and relevant experimental details within the video. However, it functions strictly as an auxiliary tool and its outputs are always subject to rigorous human validation and do not determine the final content of the benchmark.
>
> All LLM-generated outputs undergo **heavy human validation** at every stage. In the initial round, the system produced more than **4,000 questions**, but the majority were filtered out first by the visual-blind test using GPT-4o and then through multi-stage human review due to issues such as insufficient difficulty, timestamp inaccuracies, or lack of answerability. Only questions that passed strict human verification were retained in the final benchmark, ensuring correctness and grounding regardless of the LLM used in early drafts.
>
> To further demonstrate that SciVideoBench does **not** favor Gemini-2.5-Pro, we include evaluation results from **GPT-5** and **Gemini-3-Pro**, both of which outperform Gemini-2.5-Pro in quantitative reasoning. And in the case of Gemini-3-Pro, it also perform much better overall. This provides strong evidence that the benchmark does not systematically align with the strengths or biases of Gemini-2.5-Pro.
>
> | Models            | Overall | Conceptual | Hypothetical | Quantitative |
> |-------------------|---------|------------|--------------|--------------|
> | Gemini-2.5-Pro    | 64.30   | 69.73      | 67.79        | 50.61        |
> | GPT-5             | 55.70   | 61.35      | 50.65        | **55.10**    |
> | **Gemini-3-Pro**  | **70.40** | **76.22**  | **72.21**    | **58.78**    |
>
>
> ## 2. The Cognitive Complexity of Different Reasoning Types
>
> To better reflect the cognitive complexity embedded in SciVideoBench, our question generation pipeline also incorporates **explicit reasoning-chain generation** using Gemini-2.5-Pro. For each question, the model first produces a step-by-step reasoning process, which is then **manually verified by human annotators** to ensure factual correctness, alignment with the video, and consistency with scientific principles.
>
> Cognitive complexity is quantitatively reflected through two measurements:
> - the **average number of reasoning steps**
> - the **average number of reasoning words**
>
> These metrics provide a systematic view of the depth of reasoning required across different question types. The results are summarized below:
>
> | Question Type | Avg. Reasoning Steps | Avg. Reasoning Words |
> |---------------|-----------------------|------------------------|
> | **Conceptual**    | 5.58                  | 101.56                 |
> | **Hypothetical**  | 4.78                  | 91.00                  |
> | **Quantitative**  | 10.59                 | 342.63                 |
>
> These statistics demonstrate that SciVideoBench questions require multi-step reasoning, with **quantitative reasoning in particular demanding significantly deeper and longer inference chains**, aligning with research-level scientific analysis.

---

> ### Author Response · Authors · 2025-11-22
> **[2/3] General Response**
>
> ## 3. Frame Sampling Study
>
> For fairness and consistency, we follow each model’s **official default frame-sampling strategy** during evaluation. Because SciVideoBench contains visually dense experimental procedures, increasing the number of frames naturally provides **more visual evidence**, which should help models better track temporal dependencies, observe fine-grained experimental details, and reason more accurately about the underlying scientific procedures.
>
> To validate this hypothesis, we conducted additional ablation experiments on **Qwen2.5-VL (7B and 3B)** using multiple frame-sampling settings. The results below clearly show that, for both model sizes, in general, performance **drops consistently as fewer frames are provided**, confirming that SciVideoBench requires substantial visual information and is sensitive to temporal coverage.
>
> | Model & Frames | Overall  | Conceptual  | Hypothetical| Quantitative |
> | -------------- | -------- | ---------- | ------------ | ------------ |
> | Qwen2.5-VL-7B |  |  | |  |
> | **768f**  | **26.3** | **35.95**      | **28.05**        | 8.98        |
> | **128f** | 25.8 | 34.86 | 25.97 | 11.84 |
> | **64f** | 25.5 | 33.78 | 26.23 | 11.84 |
> | **32f**   | 25.4     | 35.14      | 24.42        | **12.24**        |
> | **16f** | 22.2 | 29.19 | 22.34 | 11.43 |
> | Qwen2.5-VL-3B |  |  | |  |
> | **768f**  | **22.5** | **32.43**      | **20.78**        | **10.20**        |
> | **128f** | 22.1 | 31.89 | 20.26 | **10.20** |
> | **64f** | 20.7 | 31.08 | 17.66 | 8.98 |
> | **32f** | 20.9 | 29.46 | 19.48 | **10.20** |
> | **16f** | 19.9 | 27.84 | 18.96 | 9.39 |
>
> These ablations confirm that SciVideoBench rewards models that can effectively leverage **rich temporal visual signals**, and that higher frame sampling provides clear benefits for scientific video reasoning.
>
>
> ## 4. Detailed Modality Ablation
>
> To further address the reviewer’s concern regarding modality-specific evidence, we extend our analysis by comparing **visual-blind, option-only, transcript-only, video**, and the newly added **video+paper** settings. These experiments were conducted using **GPT-4o** to isolate the contribution of each modality and to assess whether external scientific descriptions (the associated JoVE paper) provide additional advantages beyond the video alone.
>
> The updated results reveal the following:
>
> - **Option-only Guess accuracy remains close to random**, confirming that guessing is ineffective under our 10-option MCQA setting.
> - **Transcript-only QA performance outperforms visual-blind**, indicating that narration contributes useful—but insufficient—context.
> - **Video+Paper slightly improves performance over video-only**, showing that reading the accompanying research article can help in certain conceptual or hypothetical questions.
> - **However, the improvement is modest**, demonstrating that the benchmark primarily relies on *visual grounding*, not textual background knowledge.
>
> | Mode            | Overall | Conceptual | Hypothetical | Quantitative |
> |-----------------|---------|------------|--------------|--------------|
> | Option-only Guess     | 10.5   | 11.35      | 10.65        | 8.98        |
> | Visual-blind QA    | 15.80   | 14.05      | 20.00        | **11.84**        |
> | Transcript-only QA | 21.50   | 26.22      | 23.12        | **11.84**        |
> | Video QA       | 24.90   | 30.27      | 28.05        | **11.84**        |
> | Video+Paper QA | **25.80** | **30.81**  | **30.13**    | 11.43        |
>
> We also added the audio-only evaluation for Gemini-2.5-Pro below.
>
> | Modality | Overall | Conceptual | Hypothetical | Quantitative |
> |----------|---------|------------|--------------|--------------|
> | **Audio-only** | 34.90 | 38.92 | 39.22 | 22.04 |
> | **Video-only** | 64.30 | 69.73 | 67.79 | 50.61 |
> | **Audio + Video** | **67.00** | **74.32** | **67.53** | **55.10** |
>
> These results confirm that SciVideoBench questions **cannot be solved through text or reasoning priors alone**, and that the benchmark requires **genuine visual understanding**, with the accompanying paper offering only limited incremental benefit.
>
> ---
>
> We hope these expanded analyses address the reviewers’ concerns and strengthen the overall contribution of SciVideoBench. All improvements have been reflected in the updated manuscript.
>
> Sincerely,
>
> The Authors

---

### Author Response · Authors · 2025-12-03
**Rebuttal Summary for ACs and PCs**

Dear Area Chair,

Thank you very much for handling our submission. We greatly appreciate the reviewers’ detailed and constructive feedback. Across all four reviews, SciVideoBench was consistently recognized as addressing an important gap in *research-level scientific video reasoning*, with strengths noted in its novelty, annotation pipeline, domain breadth, and comprehensive model evaluation. The reviewers’ comments largely focused on requests for **additional clarification and elaboration**, such as providing more detail on the human annotation pipeline, explaining the interpretation of the human evaluation, adding modality and frame-sampling ablations, and further formalizing our characterization of scientific reasoning depth, rather than identifying flaws or errors. We have addressed each of these clarification requests in the rebuttal and outlined the corresponding updates.

## **Positive Feedback Highlighted by Reviewers**

- **Addresses a critical unmet need** (wpTT, aWZV, FZjT, 1k1D):
  All reviewers agreed that SciVideoBench fills an important gap in evaluating *research-level* scientific video reasoning, far beyond everyday or college-level video QA. They highlighted its coverage of 25+ subjects across Biology, Chemistry, Physics, and Medicine, and its high relevance to future multimodal AI for science.

- **Strong novelty and benchmark design** (wpTT, aWZV, FZjT, 1k1D):
  Reviewers recognized SciVideoBench as the first benchmark focused on *experimental videos*, integrating multimodal evidence with conceptual, hypothetical, and quantitative reasoning. They emphasized its novelty in targeting laboratory contexts rather than recognition-style tasks.

- **High-quality annotation pipeline** (wpTT, aWZV, FZjT, 1k1D):
  Reviewers praised the rigor of the multi-agent, human-in-the-loop annotation system and its strong quality control. They noted the use of specialized agents, expert verification, and precise timestamp alignment as evidence of a well-engineered dataset.

- **Comprehensive model evaluation** (wpTT, aWZV, FZjT, 1k1D):
  Reviewers appreciated the breadth of the evaluation across 35 models, including proprietary and open-source systems, CoT prompting baselines, and model-scaling comparisons.

- **Meaningful empirical insights** (wpTT, aWZV, 1k1D):
  Reviewers noted that SciVideoBench reveals significant capability gaps in existing LMMs. They valued the detailed analysis of failure modes, modality dependence, and the strong explanatory insights provided by the results.

---

> ### Author Response · Authors · 2025-12-03
> **[1/2] Rebuttal Response to Reviewers' Comments**
>
> ## **Rebuttal Response to Reviewers' Comments**
>
>
> ### **1. Annotation Process Clarity & Quality Control**
>
> Reviewers **wpTT, FZjT, and 1k1D** requested greater transparency regarding the human-in-the-loop pipeline, inter-annotator consistency, and answerability checks. In the rebuttal, we clarified that SciVideoBench uses a **multi-stage annotation workflow** involving **five PhD-level annotators**, where each question is double-reviewed and disagreements are resolved through structured expert adjudication, which provides a practical, high-fidelity substitute for formal agreement metrics. We detailed the **answerability and temporal-grounding audits**, which ensure all questions are solvable strictly from the video and align precisely with the referenced timestamps. We also clarified that **Gemini-2.5-Pro was only used for draft captioning**, not for determining final answers, and that all LLM-generated content undergoes rigorous human verification. To ensure **LLM-independence**, we introduced **GPT-4o visual-blind filtering**, confirming that questions cannot be solved through textual artifacts alone. Finally, we demonstrated that more advanced models (**GPT-5, Gemini-3-Pro**) surpass Gemini-2.5-Pro on SciVideoBench, confirming the benchmark’s **neutrality** and independence from any specific annotation-time model.
>
>
>
> ---
>
> ### **2. Human Study Design & Interpretation of the 17.4% Human Accuracy**
>
> Reviewers **aWZV, FZjT, and 1k1D** requested clearer justification of the human baseline and interpretation of the low accuracy. In the rebuttal, we clarified that the study used **three STEM graduate students per discipline**, all scientifically trained but not specialists in every sub-domain represented in SciVideoBench. Participants answered questions in a **strict closed-book setting** with only the videos, questions, and options available, and were allowed to freely replay videos during a single-session evaluation. We explained that the benchmark spans **25+ specialized research subjects**, many requiring fine-grained procedural understanding, domain-specific measurements, and multi-step scientific reasoning—skills that naturally exceed the training of non-specialist graduate students. Thus, the **17.4% accuracy reflects genuine research-level complexity**, not calibration issues. We also illustrated how substantial sub-discipline heterogeneity (e.g., organic vs. analytical chemistry, condensed-matter vs. imaging physics) makes full domain coverage unrealistic for human evaluators. Overall, the human results provide a **meaningful, intentionally challenging baseline** aligned with the benchmark’s goal of assessing advanced scientific video reasoning.
>
>
> ---
>
> ### **3. Scientific Reasoning Depth & Question Difficulty Calibration**
>
> Reviewer **wpTT** requested a clearer framework for defining research-level scientific reasoning and validating question difficulty. In the rebuttal, we clarified that SciVideoBench’s three reasoning types naturally encode different difficulty levels, with quantitative questions consistently emerging as the most challenging across all models. We introduced a **formal definition of research-level reasoning**, emphasizing skills such as interpreting real experimental procedures, performing counterfactual reasoning, analyzing quantitative measurements, and integrating multimodal scientific evidence。 These capabilities go beyond textbook-level tasks. To further substantiate this, we detailed five criteria that classify a question as research-level and reported **cognitive-complexity metrics** (average reasoning steps and words) for each reasoning type. These analyses show that all questions require multi-step reasoning, and that **quantitative items involve substantially deeper inference chains**, aligning with the benchmark’s design and supporting the observed difficulty patterns.

---

> ### Author Response · Authors · 2025-12-03
> **[2/2] Rebuttal Response to Reviewers' Comments**
>
> ### **4. Additional Ablations & Analyses**
>
> Reviewers **aWZV, FZjT, and 1k1D** requested deeper modality and frame-sampling analyses to ensure SciVideoBench cannot be solved through shortcuts. In the rebuttal, we added **audio-only, text-only, transcript-only, and visual-blind** experiments, showing that audio provides helpful context but cannot substitute for video, and that option-only accuracy remains near random, confirming **no distractor leakage**. Transcript-only performance improves over visual-blind yet remains far below video, and **quantitative questions consistently require visual inspection**, reinforcing the benchmark’s strict visual grounding. We further introduced **frame-sampling ablations** using Qwen2.5-VL (3B/7B), demonstrating that performance improves as more frames are provided, indicating the need for **rich temporal visual evidence**. Collectively, these analyses confirm that **text and audio alone are insufficient** and that SciVideoBench fundamentally requires grounded, visually intensive scientific reasoning.
>
>
> ---
>
> ### **5. Dataset Scale, Coverage, and Format**
>
> Reviewer **1k1D** raised questions about the dataset size, use of multiple-choice format, and reliance on JoVE. In the rebuttal, we clarified that the benchmark’s **1,000 rigorously validated MCQs** are on par with or larger than recent research-level video benchmarks (e.g., Video-MMMU’s 900 items), and that each SciVideoBench question requires costly timestamp alignment and multi-stage expert verification. We explained that **multiple-choice QA** was chosen over open-ended QA due to the well-known evaluation instability of free-form responses, making MCQA a **stable, reproducible, and bias-resistant** choice for scientific reasoning. The **10-option format** further reduces guessability to 10% and, through our distractor-refinement pipeline, avoids pattern leakage while maintaining option-only accuracy near random levels, yielding a highly discriminative evaluation. Finally, we noted that SciVideoBench already includes **human-verified reasoning chains**, enabling future research on rationale-based evaluation even though MCQA is used for primary scoring.
>
>
> ---
>
>
> Overall, we believe the rebuttal directly addressed the reviewers’ central concerns while further strengthening the clarity, methodological rigor, and positioning of SciVideoBench. We sincerely appreciate your time and consideration.

---

### Note · Program_Chairs · 2026-01-17
**Submission Desk Rejected by Program Chairs**

The following references in this submission do not refer to real documents and/or have major errors in bibliographic information:

 Hyeonhu Kim, Daniel Uzan, Tian Xie, et al. Accelerated discovery of stable materials using deep learning. Nature, 620:1018-1024, 2023.